# Decreasing Indian summer monsoon in northern Indian sub-continent during the last 180 years: evidence from five tree cellulose oxygen isotope chronologies

Chenxi Xu[1,2], Masaki Sano[3,4], A. P. Dimri[5], Rengaswamy Ramesh[6,7], Takeshi Nakatsuka[3], Feng Shi[1,2], Zhengtang Guo[1,2,8]

1. Key Laboratory of Cenozoic Geology and Environment, Institute of Geology and Geophysics, Chinese Academy of Sciences, Beijing 100029, China
2. CAS Center for Excellence in Life and Paleoenvironment, Beijing, 100044,China
3. Research Institute for Humanity and Nature, 457-4 Motoyama, Kamigamo, Kita-ku, Kyoto, Japan
4. Faculty of Human Sciences, Waseda University, 2-579-15 Mikajima, Tokorozawa 359-1192, Japan
5. School of Environmental Sciences, Jawaharlal Nehru University, New Delhi, India
6. Geoscience Division, Physical Research Laboratory, Navrangpura, Ahmedabad 380009, India
7. School of Earth and Planetary Sciences, National Institute of Science Education and Research, Odisha 752050, India
8. University of Chinese Academy of Sciences, Beijing, China

*Correspondence to*: Masaki Sano, (msano@aoni.waseda.jp)

**Abstract.** We have constructed a regional tree-ring cellulose oxygen isotope ($\delta^{18}O$) record for the northern Indian sub-continent based on two new records from northern India and central Nepal and three published records from northwestern India, western Nepal and Bhutan. The record spans the common interval from 1743-2008 CE. Correlation analysis reveals that the record is significantly and negatively correlated with the three regional climatic indices: All India Rainfall ($r = -0.5$, $p <0.001$, $n =138$), Indian monsoon index ($r = -0.45$, $p <0.001$, $n =51$) and the intensity of monsoonal circulation ($r = -0.42$, $p <0.001$, $n =51$). The close relationship between tree-ring cellulose $\delta^{18}O$ and the Indian summer monsoon (ISM) can be explained by oxygen isotope fractionation mechanisms. Our results indicate that the regional tree-ring cellulose $\delta^{18}O$ record is suitable for reconstructing high-resolution changes in the ISM. The record exhibits significant inter-annual and long-term variations. Inter-annual changes are closely related to the El Niño-Southern Oscillation (ENSO), which indicates that the ISM was affected by ENSO in the past. However, the ISM-ENSO relationship was not consistent over time, and it may be partly modulated by Indian Ocean sea surface temperature (SST). Long-term changes in the regional tree-ring $\delta^{18}O$ record indicate a possible trend of weakened ISM intensity since 1820. Decreasing ISM activity is also observed in various high-resolution ISM records from southwest China and Southeast Asia, and may be the result of reduced land-ocean thermal contrasts since 1820 CE.

## 1 Introduction

The Indian summer monsoon (ISM) delivers a large amount of summer precipitation to the Indian continent, and thus has a major influence on economic activity and society in this densely-populated region (Webster et al., 1998). Current research on the ISM is mainly concerned with the study of inter-annual and inter-decadal variations, using meteorological data and climate models. El Niño-Southern Oscillation (ENSO) has great influences on ISM at inter-annual time scales, and El Niño events (Warm phase of ENSO) usually produced ISM failure (Kumar et al., 1999; Kumar et al., 2006; Webster et al., 1998). North Atlantic Sea surface temperature (SST) affected ISM by modulating tropospheric temperature over Eurasia (Goswami et al., 2006; Kripalani et al., 2007). Climate model experiments indicate that there is a significant increase in mean ISM precipitation of 8% under the doubling atmospheric carbon dioxide concentration scenario (Kripalani et al., 2007) and human-influenced aerosol emissions mainly resulted in observed precipitation decrease during the second half of the 20th century (Bollasina and Ramaswamy, 2011). A good understanding of mechanisms driving ISM change on different time scales could help to predict possible changes of ISM in the future. However, the observed meteorological records are too short to assess long-term changes in ISM. Therefore, long-term proxy records of ISM are needed.

The abundance of *Globigerina bulloides* in marine sediment cores from the Arabian Sea indicated a trend of increasing ISM strength during the last 400 years (Anderson et al., 2002). However, oxygen isotopes in tree rings and ice cores from the Tibetan Plateau revealed a weakening trend ISM since 1840 or 1860 (Duan et al., 2004; Grießinger et al., 2016; Liu et al., 2014; Wernicke et al., 2015). In addition, a stalagmite oxygen isotope record from northern India indicated that the ISM experienced a 70-year pattern of variation over the last 200 years, with no clear trend (Sinha et al., 2015). Since there are spatial differences in the patterns of climate change in monsoonal areas (Sinha et al., 2011), geological records with a wide

distribution are needed. In addition, the climate proxies should be closely related to the ISM and the records need to be well-replicated and accurately dated.

Available tree-ring records are widely distributed in the Indian monsoon region (Yadav et al., 2011). The climate of the

southern Himalaya is dominated by changes in the Indian summer monsoon, and therefore the region is well suited to the study of Indian monsoon variations. The oxygen isotopic composition ($\delta^{18}$O) of tree rings is mainly controlled by the $\delta^{18}$O of precipitation and by relative humidity (Ramesh et al., 1985; Roden et al., 2000), and both are affected by the Indian summer monsoon (Vuille et al., 2005). Compared with tree-ringwidth data, tree-ring $\delta^{18}$O records are more suited to retrieving low-frequency climate signals, and therefore they have the ability to record the Indian summer monsoon (Gagen et al., 2011;

Sano et al., 2012; Sano et al., 2013). In addition, tree-ring $\delta^{18}$O is considered as a promising proxy for next phase of Past Global Changes (PAGES) 2k network not only for hydroclimate reconstruction in Asia but also for data-model comparison to understand the mechanisms of climate variability at decadal to centennial timescales.

PAGES launched 2k network that produced regional and global temperature and precipitation syntheses based on multi-

proxy and multi-record to obtain a better understanding of regional and global climate change. The ISM affected the large area of Indian continent, and a local record may not be fully representative of changes in the ISM. Therefore, we produced regional syntheses based on five tree-ring $\delta^{18}$O records from the ISM region. Two new records from northern India and central Nepal were obtained in this study, and were combined with three previously published records from northwestern India, western Nepal and Bhutan (Sano et al., 2012; Sano et al., 2013; Sano et al., 2017a). The data were integrated in order

to produce a regional tree-ring $\delta^{18}$O record which was used to reconstruct the history of the ISM during the last several hundred years, and to investigate its possible driving mechanisms on various time scales.

## 2 Materials and methods

### 2.1 Sampling sites

Five tree-ring cellulose $\delta^{18}O$ records were used to construct a regional climate signal for the southern Himalaya (Figure 1). Three records (Manali, in northwestern India; Humla, in western Nepal; and Wache, in Bhutan) were published previously (Sano et al., 2012; Sano et al., 2013; Sano et al., 2017a2017). Two tree-ring cellulose $\delta^{18}O$ chronologies were constructed in this study. Core samples for *Cedrus deodara* near Jageshwar (29°38'N, 79°51'E, 3849 m a.s.l., JG) and *Abies spectabilis* near Ganesh (28°10'N, 85°11'E, 3550 m a.s.l.) were collected in 2009 and 2001, respectively. Information about each sampling site is shown in Table 1. In general, two core samples for each tree were collected at breast height using a 5-mm diameter increment corer. The cores were air dried at room temperature for 2-3 days and the surfaces were then smoothed with sand paper to render the ring boundaries clearly visible. The ring widths of the samples were measured at a resolution of 0.01mm using a binocular microscope with a linear stage interfaced with a computer (Velmex™, Acu-Rite). Cross dating was performed in the laboratory by matching variations in ring width from all cores to determine the calendar year of each ring. Quality control was conducted using the COFECHA computer program (Holmes, 1983).

### 2.2 Cellulose extraction, isotope measurement and chronology development

Four trees near Ganesh and three trees near Jageshwar, all with relatively wide rings, were selected for oxygen isotope analysis (Figures 2 & 3). The modified plate method (Kagawa et al., 2015; Xu et al., 2011; Xu et al., 2013b), based on the chemical treatment procedure of the Jayme-Wise method (Green, 1963; Loader et al., 1997), was used to extract α-cellulose. The plate method of extracting α-cellulose directly from the wood plate rather than from individual rings can reduce the α-cellulose extraction time (Xu et al., 2011). In addition, the modified plate method can reduce the amount of sample material lost during cellulose extraction, enabling sufficient material to be obtained to enable narrow rings to be measured by isotope ratio mass spectrometer (Xu et al., 2013b). There is no statistically significant difference between tree-ring $\delta^{18}O$ values

obtained by the plate and conventional methods (Kagawa et al., 2015; Xu et al., 2013b). For the samples from the Humla site, every annual ring of five individual trees was split with a scalpel, and then the shavings were pooled for each year and subjected to cellulose extraction for each year (Sano et al., 2012).

Cellulose samples (sample weight, 120-260 μg) were wrapped in silver foil, and tree-ring cellulose oxygen isotope ratios ($^{18}O/^{16}O$) were measured using an isotope ratio mass spectrometer (Delta V Advantage, Thermo Scientific) interfaced with a pyrolysis-type high-temperature conversion elemental analyzer (TC/EA, Thermo Scientific) at the Research Institute for Humanity and Nature, Japan. Cellulose $\delta^{18}O$ values were calculated by comparison with Merck cellulose (laboratory working standard), which was inserted after every eight tree samples during the measurements. Oxygen isotope results are

presented using the δ notation as the per mil (‰) deviation from Vienna Standard Mean Ocean Water (VSMOW): $\delta^{18}O =$ $[(R_{sample}/R_{standard}) - 1] \times 1000$, where $R_{sample}$ and $R_{standard}$ are the $^{18}O/^{16}O$ ratios of the sample and standard, respectively. The analytical uncertainty for repeated measurements of Merck cellulose was approximately ±0.15‰.

The local chronology was produced by averaging all the individual series for a given site. The 95% confidence intervals

(±1.96σ) for each year of the local chronology were calculated using annual $\delta^{18}O$ values of individual trees. It should be noted that the confidence intervals for the tree-ring chronology from Hulma, western Nepal cannot be calculated because the chronology was produced by pooling method, and therefore only single $\delta^{18}O$ value was measured for a year.

A regional tree-ring $\delta^{18}O$ chronology was produced using the two tree-ring $\delta^{18}O$ records from the JG and Ganesh sites and

the three published tree-ring $\delta^{18}O$ records from the Manali, Humla and Wache sites. Specifically, every $\delta^{18}O$ series from the five sites was individually normalized over the period 1951-2000 CE, and then the resulting series for each site were averaged to produce a site chronology. Finally, all the site chronologies were averaged to develop the regional Himalayan

$\delta^{18}O$ record (H5 $\delta^{18}O$ record) for the entire period (Figure 4f). The 95% confidence intervals for the regional chronology were computed using annual $\delta^{18}O$ anomalies of the five site chronologies.

*2.3 Climate analyses and Statistical Analysis*

In the northern Indian subcontinent, the monsoon season is from June to September. The summer monsoon season supplies 78% and 83% of the annual precipitation for Kathmandu and New Delhi, respectively. The Indian monsoon index (IMI) (Wang et al., 2001, http://apdrc.soest.hawaii.edu/projects/monsoon/definition.html), the intensity of monsoon circulation (Webster and Yang, 1992) and All India Rainfall (AIR, obtained from the Indian Institute of Tropical Meteorology, Pune, India, Mooley et al., 2016) were selected as proxies for the Indian summer monsoon in order to investigate the relationship

between tree-ring cellulose $\delta^{18}O$ variations and the monsoon. In addition, we used the Royal Netherlands Meteorological Institute Climate Explorer (http://www.knmi.nl/) to determine spatial correlations between tree-ring cellulose $\delta^{18}O$, precipitation (GPCC V7, Schneider et al., 2015) and SST values obtained from the National Climatic Data Center ERSST v4 data set (Huang et al., 2015). Two reconstructed ENSO indices (Emile-Geay et al., 2013; McGregor et al., 2010) based on paleoclimate records (tree-ring, coral, sediment and ice core) were used to evaluate the ISM-ENSO relationship in the past.

Temperature reconstructions for the Indian Ocean (Tierney et al., 2015) and the Tibetan Plateau (Cook et al., 2013; Shi et al., 2015; Wang et al., 2015), spanning the last 400 years, were used to obtain a record of the history of land-ocean thermal contrast. The land-ocean thermal contrast is equal to the land temperature minus the ocean temperature. Because uncertainty of the temperature reconstruction was provided as RMSE (Root Mean Square Error) in the previous studies, the uncertainty range for the land-ocean contrast was estimated as the sum of RMSE of land temperature and Indian Ocean SST for each

20   year. The low-frequency signals together with uncertainties (>100 years) for the tree-ring $\delta^{18}O$ chronology and the land-ocean contrast were extracted by using a low-pass filter. The software "kSpectra Toolkit" was employed to calculate power spectrum of the regional tree-ring $\delta^{18}O$ chronology.

**3 Results and discussion**

*3.1 Tree-ring $\delta^{18}O$ variations in the southern Himalaya and a regional tree-ring $\delta^{18}O$ record*

The oxygen isotopes of four individuals of *Abies spectabilis* in Ganesh (GE, central Nepal) and three individuals of *Cedrus*

*deodara* in Jageshwar (JG, northern India) were measured for the interval from 1801-2000 CE and 1621-2008 CE, respectively. Individual tree-ring $\delta^{18}O$ time series from four cores from central Nepal are shown in Figure 2a. The mean values (standard deviations) of the $\delta^{18}O$ time series from 224c, 233b, 235b, and 226a are 23.09‰ (1.22‰), 22.66‰ (1.27‰), 21.87‰ (1.12‰), and 22.94‰ (1.42‰), respectively, from 1901-2000 CE. The inter-tree differences in $\delta^{18}O$ values are small. The $\delta^{18}O$ values of the three cores exhibit peaks in 1813. The mean inter-series correlations (Rbar) among the cores

range from 0.56-0.78 (Figure 2c), based on a 50-year window over the interval from 1801-2000 CE.

Three tree-ring $\delta^{18}O$ time series from northern India (JG) are shown in Figure 3a. The mean values (standard deviations) of the $\delta^{18}O$ time series from 101c, 102c, and 103a are 30.11‰ (1.49‰), 29.7‰ (1.62‰) and 29.47‰ (1.53‰), respectively, over the interval from 1694-2008 CE. Three tree-ring $\delta^{18}O$ time series in JG exhibit a consistent pattern of variations. The

15 mean inter-series correlations (Rbar) among the cores range from 0.61-0.78 (Figure 3c), based on a 50-year window over the interval from 1621-2008 CE.

In northern Indian sub-continent, three long-term tree-ring $\delta^{18}O$ chronologies from northwestern India, western Nepal and Bhutan have been built up in previous studies (Sano et al., 2012; Sano et al., 2013; Sano et al., 2017a, Figure 4). Two tree-

20 ring $\delta^{18}O$ chronologies in this study and three tree-ring $\delta^{18}O$ chronologies in previous studies originate in monsoonal area (Figure 1). Three tree-ring $\delta^{18}O$ chronologies in northwestern India, western Nepal and Bhutan were controlled by monsoon season rainfall or PDSI (Sano et al., 2012; Sano et al., 2013; Sano et al., 2017a, Table 1), and the two new tree-ring $\delta^{18}O$

records obtained in the present study (JG and Ganesh) are negatively correlated with June-September PDSI in northern India (Table 1). In addition, the five tree-ring $\delta^{18}O$ records for the Himalayan region are significantly correlated with one another at inter-annual time scale during the common period, and in most cases 31-year running averages of five tree-ring $\delta^{18}O$ chronologies show significant correlations at multi-decadal time scale (Table 2). These results indicate that five tree-ring $\delta^{18}O$ records reflect a common controlling factor that may be related to regional climate. Figure 4f represents the regional tree-ring $\delta^{18}O$ chronology produced using the five local tree-ring chronologies. Because only one local chronology (JG) spans an interval prior to 1742 CE, we focus only on the interval from 1743-2008 CE in the following analysis.

*3.2 Climatic signals in the regional tree-ring $\delta^{18}O$ chronology*

We assessed the potential of the H5 $\delta^{18}O$ record as an indicator of past monsoon changes by correlating it with All India Rainfall (AIR), the Indian Monsoon Index (IMI) and the intensity of monsoon circulation. The results revealed a significant negative correlation with AIR ($r = -0.5$, $p <0.001$, $n =138$), IMI ($r = -0.45$, $p <0.001$, $n =51$) and the intensity of the monsoon circulation ($r = -0.42$, $p <0.001$, $n =51$) (Figure 5). In addition, the results of spatial correlation analyses reveal that the H5 $\delta^{18}O$ record is negatively correlated with gridded June-September precipitation in northern India and Nepal (Figure 6). These findings indicate that the H5 $\delta^{18}O$ record is capable of reflecting ISM changes from a statistical perspective.

Tree-ring $\delta^{18}O$ has a close relationship with the ISM based on tree-ring cellulose oxygen isotope fractionation model. Precipitation $\delta^{18}O$ and relative humidity are the two main factors controlling tree-ring $\delta^{18}O$ (Roden et al., 2000), and both are related to ISM changes in the monsoonal area. There is a negative correlation between the ISM and precipitation $\delta^{18}O$ in the monsoonal area (Vuille et al., 2005; Yang et al., 2016). Asian summer monsoon affects the $\delta^{18}O$ of precipitation through the amount effect (Cai and Tian, 2016; Dansgaard, 1964; Lekshmy et al., 2015). A stronger summer monsoon usually brings more summer rainfall to the southern Himalaya. The removal of the heavier isotopes during the condensation process results

in the oxygen isotopic depletion of the water vapor. The greater the total amount of precipitation, and the stronger the convection, the more the oxygen isotopic composition of the rainwater is affected by depletion (Lekshmy et al., 2014; Vuille et al., 2003), and this signal is reflected in tree-ring $\delta^{18}$O values. In addition, monsoon-related factors (e.g. upstream rainout process) other than the ''amount effect'' may affect precipitation $\delta^{18}$O significantly (Vuille et al., 2005). On the other hand, a stronger ISM leads to higher relative humidity, and a lower re-evaporation rate for rainfall or a reduced evaporation of leaf water in trees, resulting in less enriched tree-ring $\delta^{18}$O values (Risi et al., 2008; Roden et al., 2000).

*3.3 Interannual variability of the ISM inferred from the regional tree-ring $\delta^{18}$O record*

The results of spectral analysis using the multi-taper method (Mann and Lees, 1996) indicate that the H5 regional tree-ring $\delta^{18}$O record contains several high-frequency periodicities (4 and 5 years), as well as lower frequency periodicities at a confidence level greater than 99% (Figure 7). This indicates that interannual and long-term variability of the ISM was dominant characteristic feature during the last several hundred years. The interannual variability (4-5 years) of the H5 record is similar to that of ENSO, suggesting a possible relationship (Mason, 2001). The spatial correlation between the H5 record and SST also reveals a close relationship between the ISM and ENSO (Figure 8). Other high-resolution ISM-related records from monsoonal Asia also exhibit similar interannual periodicities (Sun et al., 2016; Xu et al., 2013a; Yadava and Ramesh, 2007). In addition, meteorological data indicates that ENSO has had a significant influence on changes in the ISM change (Kumar et al., 1999; Webster et al., 1998).

However, observational data indicate that the ENSO-ISM relationship is not consistent over time because of the southeastward shift of the descending limb of the Walker circulation and the varying monsoonal impact of the different patterns of El Niño (Kumar et al., 1999; Kumar et al., 2006). Thirty-one-year running correlations between the H5 regional tree-ring $\delta^{18}$O record and two reconstructed ENSO indices (Emile-Geay et al., 2013; Figure 9a red line; McGregor et al.,

2010, Figure 9a blue line) shared similar variations, indicating that this type of unstable ISM-ENSO relationship occurred during the last 250 years. It should be noted that these two ENSO reconstructions, which shared similar proxies such as tree-ring data from southwest United States and Mexico, were highly correlated with each other. The reason causing the unstable ENSO-ISM relationship may be responsible for the two different patterns of El Niño (developed in eastern-Pacific or central-Pacific) yielding different monsoon impacts (Kumar et al., 2006). In addition, other factors, such as Indian Ocean SST, also influence the ISM-ENSO relationship (Ashok et al., 2001; Abram et al., 2008; Wu and Kirtman, 2004).

Most proxy-based ENSO reconstructions focused on canonical El Niño events (eastern-Pacific El Niño) that are characterized by unusually warm sea surface temperatures (SST) in the eastern equatorial Pacific (Gergis and Fowler, 2009; Li et al., 2011; McGregor et al., 2010); while a different type of El Niño (central-Pacific El Niño) is characterized by warm SSTs in the central Pacific, flanked by cooler SSTs to the west and east. The latter is termed El Niño Modoki or the central-Pacific El Niño (Ashok et al., 2007; Kao and Yu, 2009; Yeh et al., 2009), and it has a different effect on the ISM (Kumar et al., 2006). In order to characterize in detail the relationship between the ISM and the two types of ENSO during the last several hundred years, spatial configuration of the SST anomalies reconstruction based on long-term coral-based records from the tropical eastern and central Pacific are needed. However, such high-resolution, continuous and robust SST reconstructions are scarce. Even in the equatorial Pacific 'centre of action' (COA) of ENSO, the COA SST reconstruction is not considered robust prior to 1850 CE (Wilson et al., 2010). A new eastern Pacific SST record for the last 400 years is not reliable during the interval from 1635-1702 CE and 1840-1885 CE (Tierney et al., 2015). In addition to ENSO's influences, Indian Ocean SST changes also have significant influences on ISM (Ashok et al., 2001), which was evident by spatial correlations between the H5 tree-ring $\delta^{18}$O chronology and SST (Figure 8). Indian Ocean SST changes that were shown to be strongly related to ENSO changes also affect ISM-ENSO relationships (Wu and Kirtman, 2004). Numerical simulations showed that ISM-ENSO relationship reversed when Indian Ocean was decoupled from the atmosphere (Wu and Kirtman,

2004). Thirty-one-year running correlations between Indian Ocean SST reconstruction (Tierney et al., 2015) with two ENSO indices reconstruction were used to evaluate the relationship between Indian Ocean SST and ENSO during the last 250 years (Figure 9b), which indicated that Indian Ocean SST showed consistent change with ENSO during most of the period but decoupled with ENSO variability during the period of 1820–1860. The ISM-ENSO relationship was weak or reversed when

Indian Ocean SST show negative correlations with ENSO. Besides, observed data and atmospheric general circulation model result revealed that ENSO's influences on ISM was reduced when a strong positive Indian Ocean Dipole (IOD) event simultaneously occurs with El Niño, and ISM-ENSO correlation is low when IOD-ISM correlation is high (Ashok et al., 2001; 2004). The weak ISM-ENSO correlation may be related to enhanced IOD-ISM correlation during the past 250 years (Figure 9a). However, absence of long-term IOD reconstruction impeded the investigation on ISM-ENSO and ISM-IOD

relationship in the past. The future availability of longer, annually resolved marine records that provide spatial configuration of SSTs in the tropical Pacific and Indian Ocean will improve our understanding of the relationship between the ISM and the two types of ENSO/Indian Ocean SST.

*3.4 Long-term of the ISM inferred from the regional tree-ring $\delta^{18}O$ record*

The long-term changes of regional tree-ring $\delta^{18}O$ record exhibits a decreasing trend from 1743 to 1820 CE and an increasing trend since 1820 CE (Figure 10c), which may indicate a weakening trend of the ISM during the interval from 1820-2000 CE. A reduction in the monsoon precipitation/relative humidity of the ISM in the last 200 years is also evident in other areas influenced by the ISM. Maar lake sediments in Myanmar exhibit a decreasing trend of monsoonal rainfall since 1840 CE (Sun et al., 2016); a tree-ring $\delta^{18}O$ record from southeast Asia exhibits a drying trend since 1800 CE (Xu et al., 2013a); a

stalagmite $\delta^{18}O$ record from southwest China reveals an overall decreasing trend in monsoon precipitation since 1760 CE (Tan et al., 2016); and tree-ring $\delta^{18}O$ and maar lake records in southwest China indicate reduced monsoon precipitation/relative humidity/cloud cover since 1840 or 1860 CE (Chu et al., 2011; Grießinger et al., 2016; Liu et al., 2014;

Wernicke et al., 2015; Xu et al., 2012). Monsoon precipitation in northwestern India shows a significant decreasing trend during the period of 1866-2006 (Bhutiyani et al., 2010).

However, in contrast, marine sediment records from the Western and Southeastern Arabian Sea exhibit an increasing trend of ISM strength over the last four centuries (Anderson et al., 2002; Chauhan et al., 2010). A recent study indicated that the contrasting trends in the ISM during the last several hundred years observed in geological records resulted from the different behavior of the Bay of Bengal branch and Arabian Sea branch of the ISM (Tan et al., 2016), and the Bay of Bengal branch of ISM weakened while intensity of Arabian Sea branch of the ISM increased during the last 200 years. However, the tree-ring $\delta^{18}O$ record in northwestern India, influenced significantly by the Arabian Sea branch of the ISM, exhibits a drying trend since 1950 CE (Sano et al., 2017a), which does not support the idea of a strengthening Arabian Sea branch of the ISM (Anderson et al., 2002). Moreover, there are no calibrated radiocarbon dates for the last 300 years for the two records from the Arabian Sea (Anderson et al., 2002; Chauhan et al., 2010). We suggest that further high-resolution and well-dated ISM records from western India are needed to improve our understanding of the behavior of the ISM. Although reconstructed All India monsoon rainfall does not show a significant decreasing trend during the period of 1813-2005 (Sontakke et al., 2008), only four stations have data extending back to 1826 CE, and are located in central or south India. Monsoon season drying trend in the Himalaya revealed by the H5 regional tree-ring $\delta^{18}O$ record may indicate that inland areas appear to be particularly sensitive to the weakening of monsoon circulation, as indicated by Sano et al. (2013).

The H5 record suggests a decreasing trend of ISM strength, which is supported by most of the other well-dated and high-resolution ISM records in ISM margin areas (Liu et al., 2014; Sun et al., 2016; Xu et al., 2012; Xu et al., 2013a). A previous study has indicated that solar irradiance has a significant influence on the ISM on multi-decadal to centennial timescales, and that reduced solar output is correlated with weaker ISM winds (Gupta et al., 2005). However, solar irradiance has increased

since 1810-1820 CE (Bard et al., 2000; Lean et al., 1995) and therefore it cannot be the main reason for the weaker ISM since 1820 CE. Atmospheric $CO_2$ content is another forcing factor for the ISM, with higher atmospheric $CO_2$ content resulting in a stronger ISM (Kripalani et al., 2007; Meehl and Washington, 1993). Thus, the increased atmospheric $CO_2$ content during the last 200 years is unlikely to be the reason for the weakened ISM.

Firm evidence of a weakening land-ocean thermal gradient over South Asia has been shown using long-term observations and model experiments by Roxy et al (2015). Rapid warming in the Indian Ocean and relatively subdued warming over the Indian subcontinent both contribute to the weakening land-ocean gradient, and thereby reduce amounts of precipitation over parts of South Asia (Roxy et al., 2015). The history of land-ocean thermal contrasts is reconstructed based on temperature

differences between the Tibetan Plateau and the Indian Ocean (Figure 10a), and centennial variations of land-ocean thermal contrasts are shown in Figure 10b. Three reconstructed land-ocean thermal contrasts showed a decreasing trend since the early 19th century (Figure 10b), and the H5 record exhibits a similar pattern of changes on a centennial scale (Figure 10c). The decreasing land-ocean thermal contrast since the early 19th century has potentially resulted in a weaker ISM, which is consistent with the increasing trend of the H5 record since 1820 CE. It should be noted, however, while the long-term trends

are overall consistent between our regional tree-ring chronology and the land-ocean thermal contrasts, possible propagation of uncertainty for the long-term land-ocean gradient data should be kept in mind for interpretation. Aerosol emissions may be another reason to cause weakened ISM. Because the aerosol-induced differential cooling of the source and nonsource regions resulted in not only reduced local land-ocean surface thermal contrast but also weaken large-scale meridional atmospheric temperature gradients, both of which caused weakening Indian summer monsoon circulation (Bollasina et al.,

2011; Cowan and Cai, 2011). Long-term aerosol emissions record is needed to evaluate aerosol emission's influences on ISM in the past.

**4 Conclusions**

We have combined three published tree-ring cellulose $\delta^{18}O$ records (from northwestern India, western Nepal and Bhutan) with two new tree-ring cellulose $\delta^{18}O$ records (from northern India and central Nepal) to produce a regional record (H5) for the northern Indian sub-continent for the interval from 1743-2008 CE. This record is significantly and negatively correlated

with All India Rainfall ($r$ = -0.5, $p$ <0.001, $n$ =138), the Indian monsoon index ($r$ = -0.45, $p$ <0.001, $n$ =51) and the intensity of the monsoon circulation ($r$ = -0.42, $p$ <0.001, $n$ =51). Spatial correlation analysis indicates that the H5 record is negatively correlated with June-September precipitation in the northern Indian sub-continent. The Indian summer monsoon (ISM) controls the tree-ring cellulose $\delta^{18}O$ record via its effects on the $\delta^{18}O$ of precipitation and relative humidity. Based on the observed statistical relationships and the physical mechanisms linking variations in tree-ring $\delta^{18}O$ and the ISM, regional tree-

ring cellulose $\delta^{18}O$ chronology in the northern Indian sub-continent is a suitable high-resolution proxy for past ISM changes.

Significant correlations between inter-annual changes of tree-ring $\delta^{18}O$ and ENSO indices indicate that the ISM was affected by ENSO. However, the ISM-ENSO relationship was not consistent in the past, and may be affected by different type of ENSO and Indian Ocean SST. A robust, high-resolution and continuous SST spatial reconstruction from the 'centre of action'

area of the Pacific and Indian Ocean would shed more light on this relationship. Long-term variations in the H5 record may reveal a trend of weakened ISM intensity since 1820 CE, which is also evident in various high-resolution ISM records from southwest China and Southeast Asia. Reduced land-ocean contrasts since 1820 CE, together with increased anthropogenic aerosol emissions during the last hundred years, may have contributed to the weakened ISM.

Data availability:

The tree-ring cellulose oxygen isotope data in this paper are available from NOAA Paleoclimatology Datasets (https://www.ncdc.noaa.gov/data-access/paleoclimatology-data). The Hulma $\delta^{18}$O chronology should be available at this link:

https://www.ncdc.noaa.gov/paleo/study/22547 (Sano et al., 2017b), which was described by Sano et al. (2012).; The Wache $\delta^{18}$O chronology should be available at this link: https://www.ncdc.noaa.gov/paleo/study/22548 (Sano et al., 2017c), which was described by Sano et al. (2013); The Manali $\delta^{18}$O chronology should be available at this link: https://www.ncdc.noaa.gov/paleo/study/22549 (Sano et al., 2017d), which was described by Sano et al. (2017a); The JG $\delta^{18}$O chronology should be available at this link: https://www.ncdc.noaa.gov/paleo/study/22550 (Xu et al., 2017a); The Ganesh $\delta^{18}$O chronology should be available at this link: https://www.ncdc.noaa.gov/paleo/study/22551 (Xu et al., 2017b). The Regional tree-ring cellulose$\delta^{18}$O chronology should be available at this link: https://www.ncdc.noaa.gov/paleo/study/23830 (Xu et al., 2017c).

**Acknowledgments:**

This work was jointly funded by the Ministry of Science and Technology of the People's Republic of China (Grant No. 2016YFA0600502), the Chinese Academy of Sciences (CAS) Pioneer Hundred Talents Program, the National Natural Science Foundation of China (Grant No. 41672179, 41630529 and 41430531), an environmental research grant from the Sumitomo Foundation, Japan, a research grant from the Research Institute of Humanity and Nature, Kyoto, Japan, and Grant in-Aid for scientific research from the Japan Society for the Promotion of Science (23-10262, and 17H01621). Indian Space Research Organization's Geosphere Biosphere Programme supported RR and APD. This study was conducted in the framework of the Past Global Changes (PAGES) Asia2k programme. We deeply appreciate the helpful comments from three anonymous reviewers, the editor and the group members of SPATIAL laboratory at the University of Utah to improve the manuscript.

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

Table 1. Tree-ring cellulose oxygen isotope data sets used in this study

| No. | Sample ID | Location | Period | Tree species | Mean | Climatic response of tree-ring $\delta^{18}O$ | Data source | Data Citations |
|---|---|---|---|---|---|---|---|---|
| 1 | Manali | 32°13′N, 77°13′E, 2700 masl, India | 1768-2008 | *Abies pindrow* | 29.97‰ | Regional JJAS PDSI $r$ =-0.67 | Sano et al., 2017 | Sano e al., 2017d |
| 2 | JG | 29°38′N, 79°51′E, 3849 masl, India | 1621-2008 | *Cedrus deodara* | 30.39‰ | Regional JJAS PDSI $r$ =-0.50 | This study | Xu et al 2017a |
| 3 | Hulma | 29°51′N, 81°56′E, 3850 masl, Nepal | 1778–2000 | *Abies spectabilis* | 25.94‰ | Regional JJAS PDSI $r$ =-0.73 | Sano et al., 2012 | Sano et al 2017b |
| 4 | | 28°10′N, 85°11′E, | 1801- | *Abies* | | Regional JJAS PDSI | This study | Xu et al |

| | | | | | | | |
|---|---|---|---|---|---|---|---|
| Ganesh | 3550 masl, Nepal | 2000 | *spectabilis* | 23.01‰ | r =-0.55 | | 2017b |
| 5 | 27°59′N, 90°00′E, 3500 masl, Bhutan | 1743-2011 | *Larix griffithii* | 19.38‰ | Regional JJAS PDSI r =-0.59 | Sano et al., 2013 | Sano et al 2017c |
| Wache | | | | | | | |

Table 2: Correlation coefficients between the tree-ring $\delta^{18}$O records from different sampling locations

| *R* (annual) | Manali | JG | Hulma | Ganesh |
|---|---|---|---|---|
| JG | 0.50* | | | |
| Hulma | 0.52* | 0.51* | | |
| Ganesh | 0.47* | 0.66* | 0.61* | |
| Wache | 0.23* | 0.26* | 0.37* | 0.52* |

| *R* (multi-decadal) | Manali | JG | Hulma | Ganesh |
|---|---|---|---|---|
| JG | 0.36* | | | |
| Hulma | 0.37* | 0.64* | | |
| Ganesh | -0.03 | 0.94* | 0.66* | |

| Wache | 0.11 | 0.39* | 0.38* | 0.70* |

*p<0.01

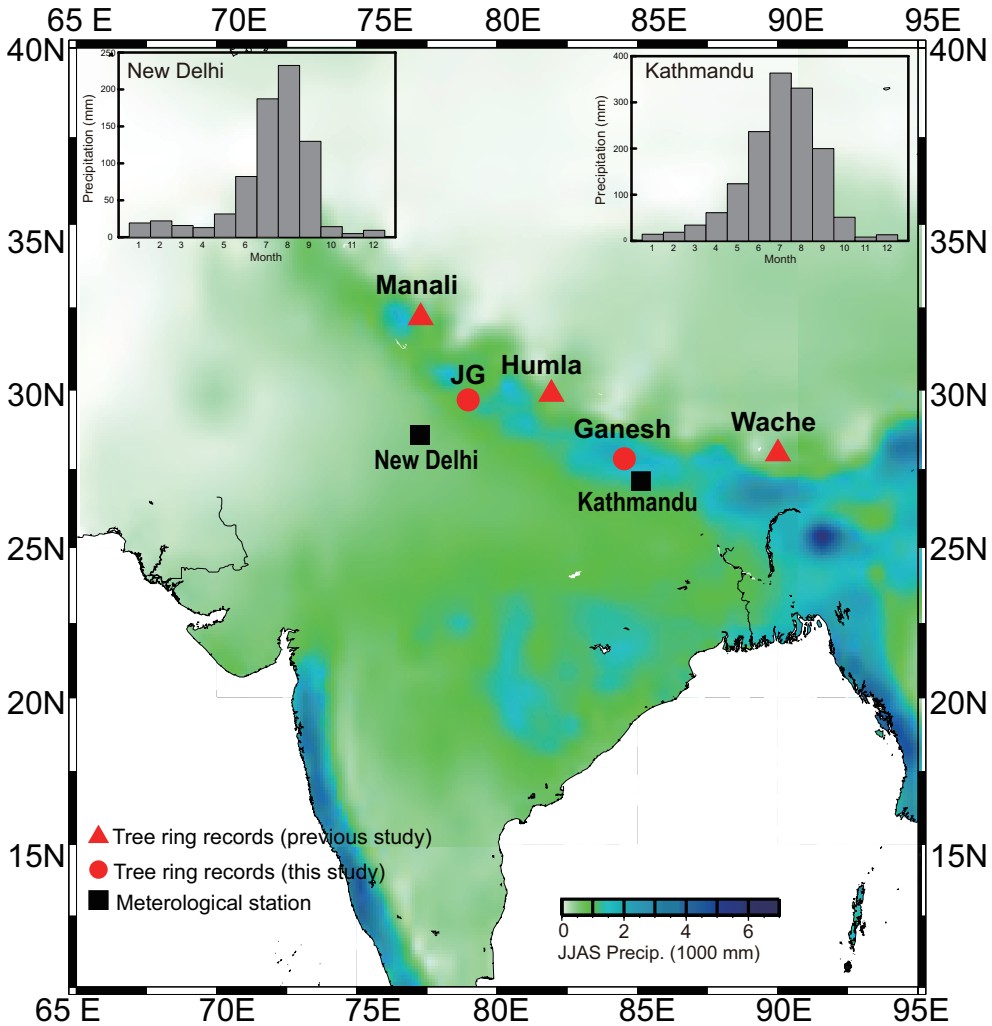

Figure 1. Map of the subcontinent showing tree-ring sites and color coded climatological monsoon precipitation from June to

September. Insets show climatology of monthly precipitation at Kathmandu and New Delhi.

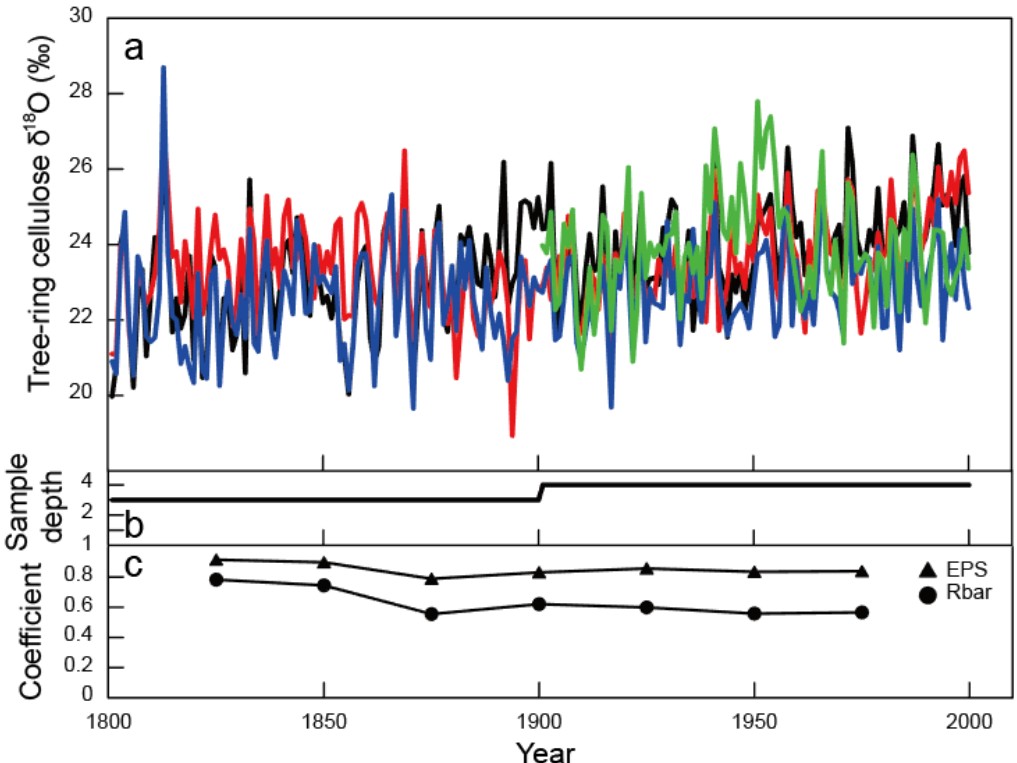

Figure 2. a: Tree-ring δ¹⁸O series of four individual trees: b: sample size, c: running EPS and Rbar statistics used 50-year windows and a 25-year lag for samples near Ganesh, Nepal.

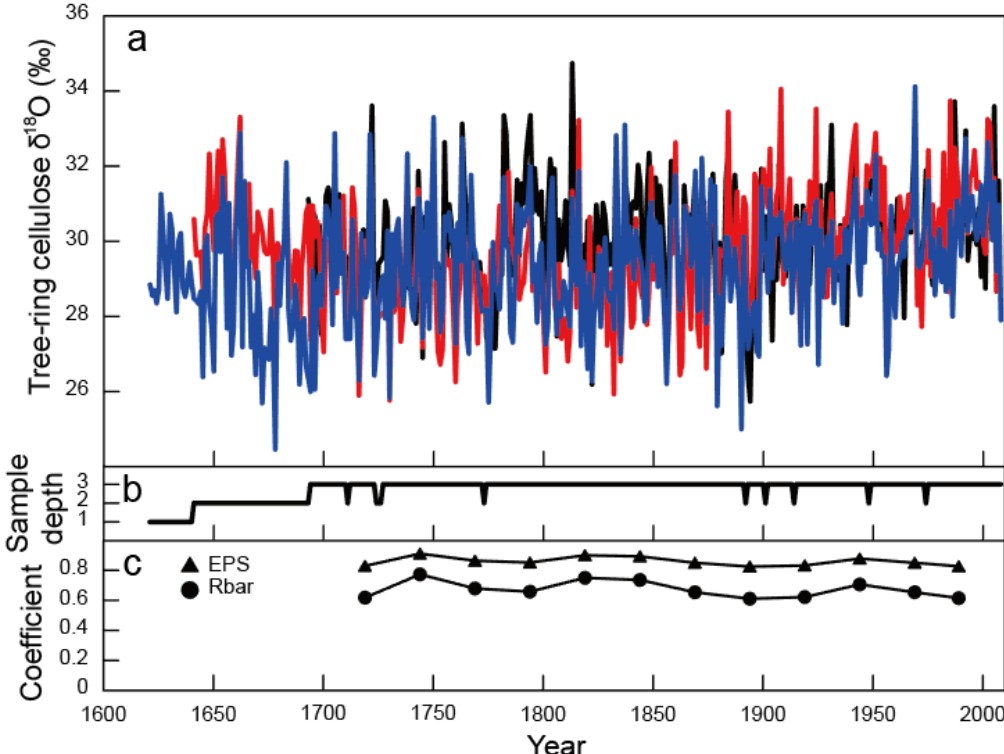

Figure 3. a: Tree-ring $\delta^{18}O$ series for three individual trees: b: sample size, c: running EPS and Rbar statistics using 50-year windows and a 25-year lag for samples near Jageshwar, India.

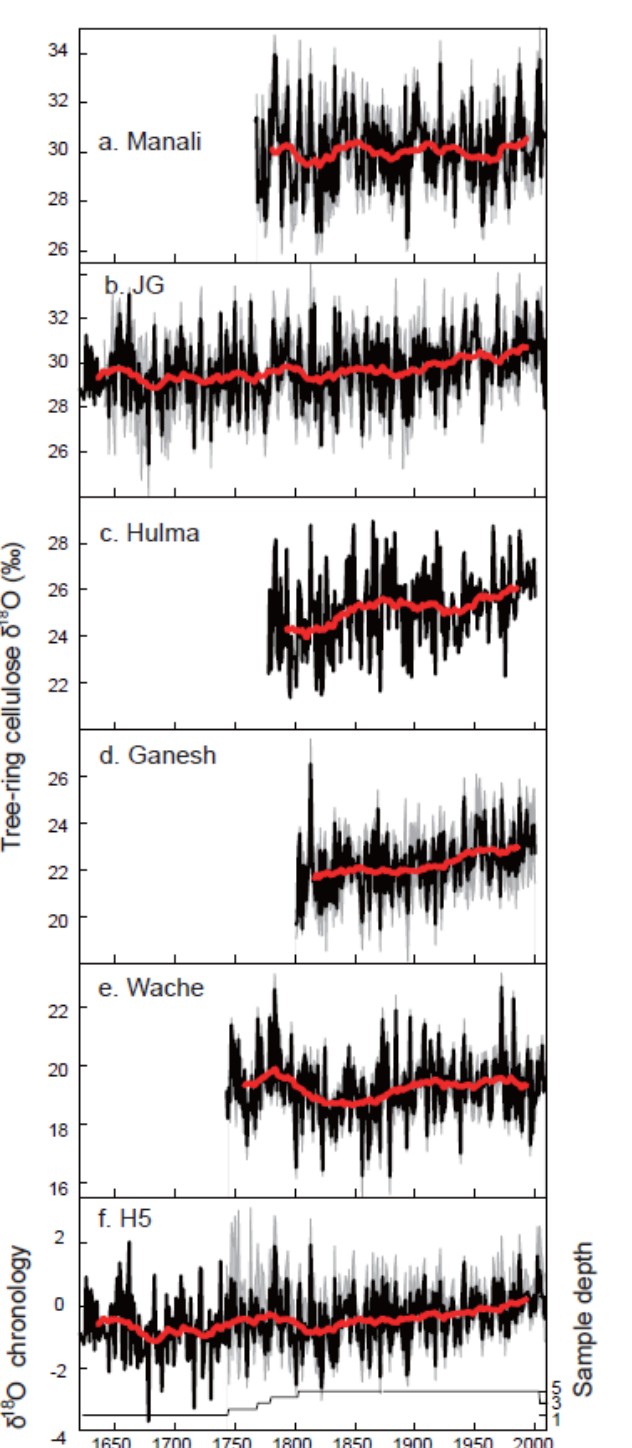

Figure 4: Tree-ring oxygen isotope chronologies from five sites (a-e) and the regional tree-ring oxygen isotope chronology (f). (black line: mean values for all samples; red line: 31-year running average for the chronology; gray shadows: the 95% confidence intervals for the local and regional chronologies.)

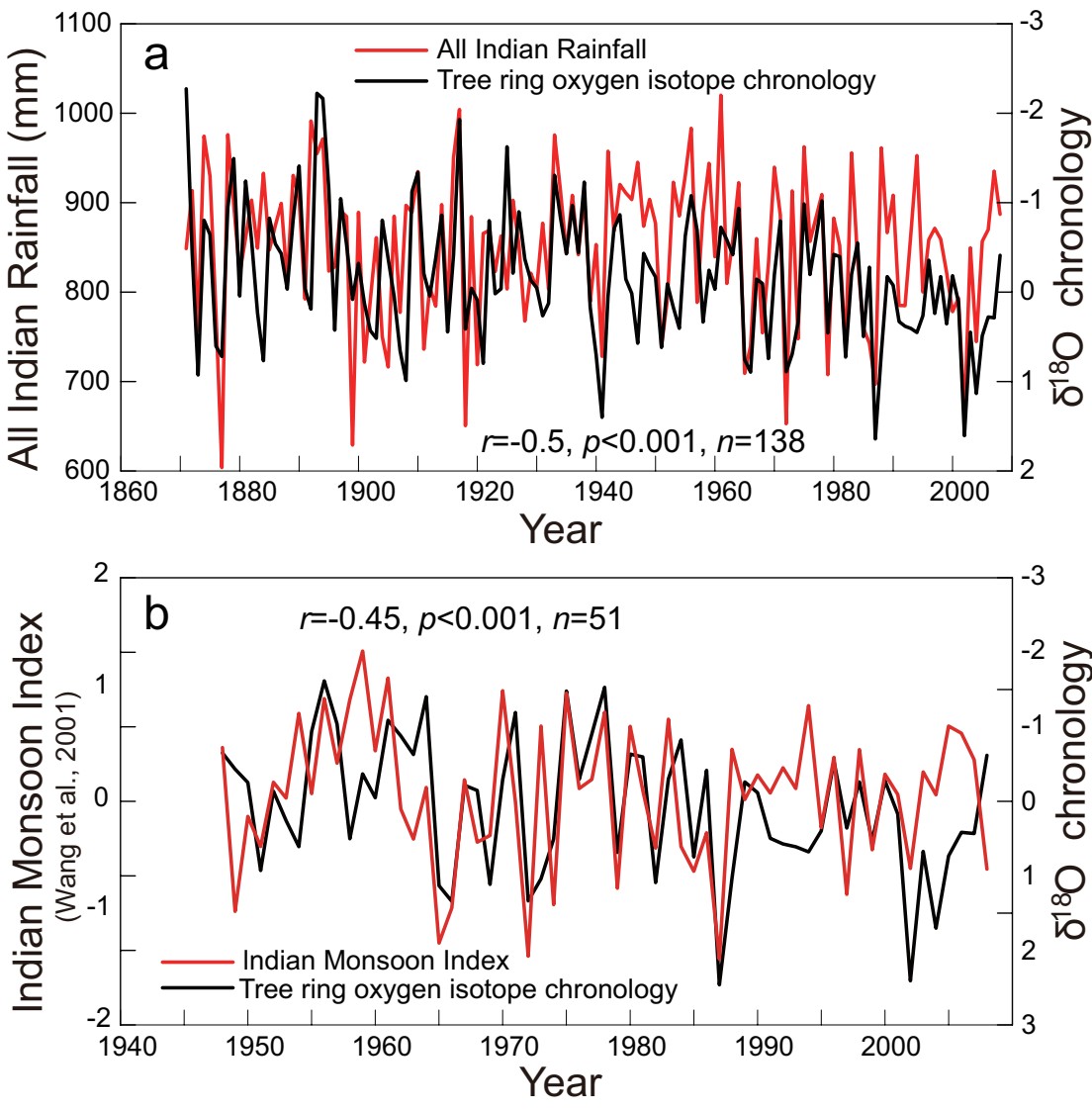

Figure 5. Comparison of the H5 regional tree-ring $\delta^{18}$O chronology with the All India Rainfall (a) and Indian Monsoon Index (b).

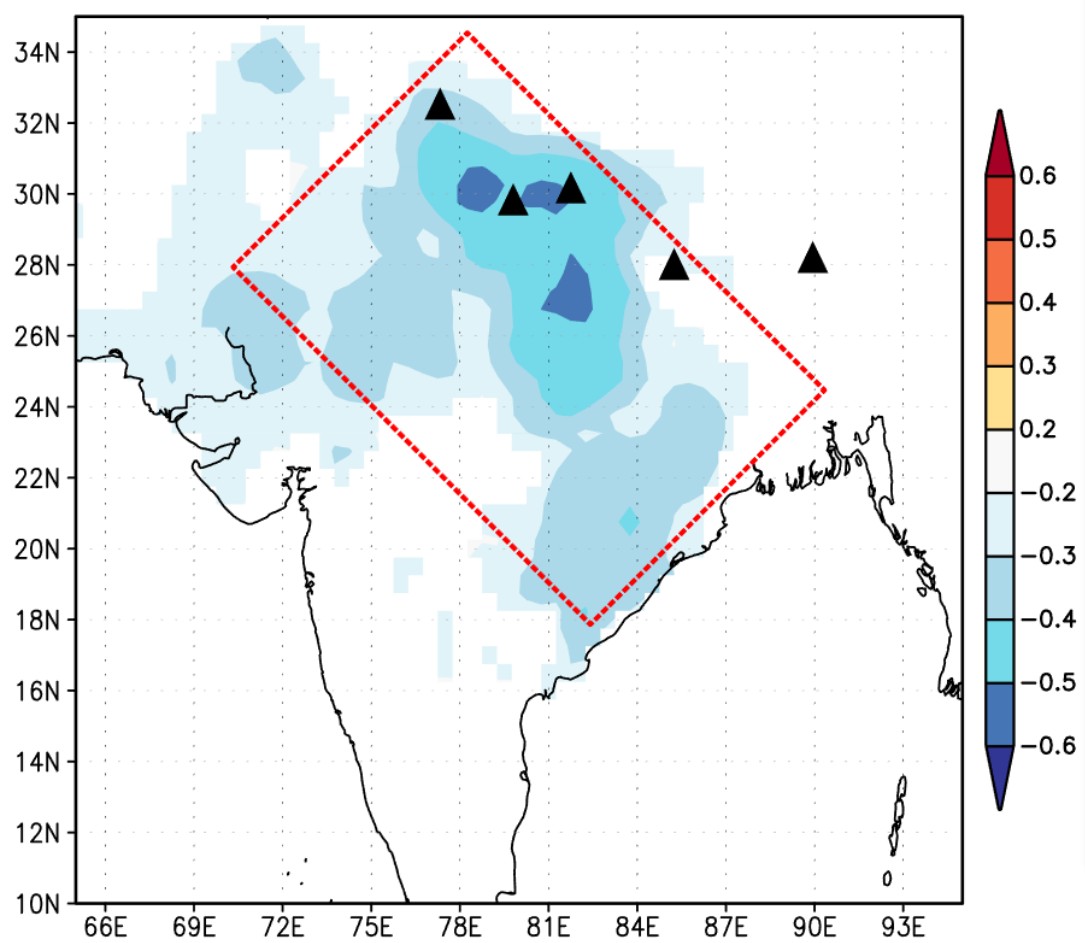

Figure 6. Spatial correlations between the H5 regional tree-ring $\delta^{18}$O record with June-September precipitation from GPCC V7 over interval from 1901-2008 CE. Only correlations significant at the 95% level are shown.

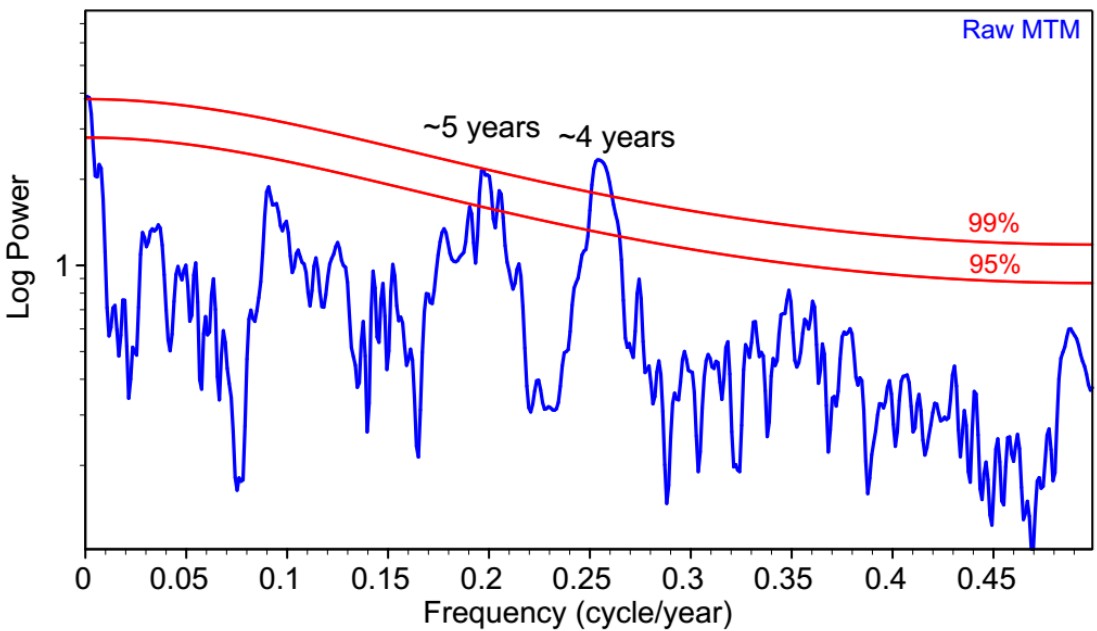

Figure 7. Multi-taper power spectra for the H5 regional tree-ring $\delta^{18}$O record.

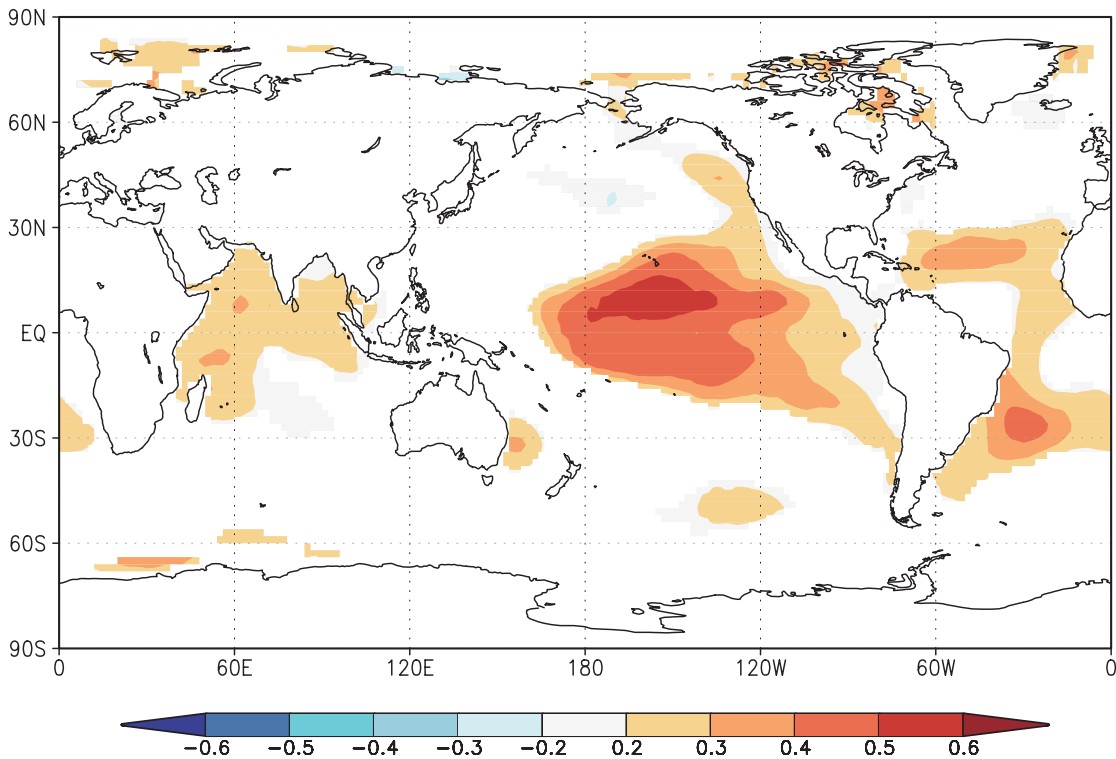

Figure 8. Spatial correlations between the H5 regional tree-ring $\delta^{18}$O record with May-September SST over the interval from 1871-2008 CE. Only correlations significant at the 95% level are shown.

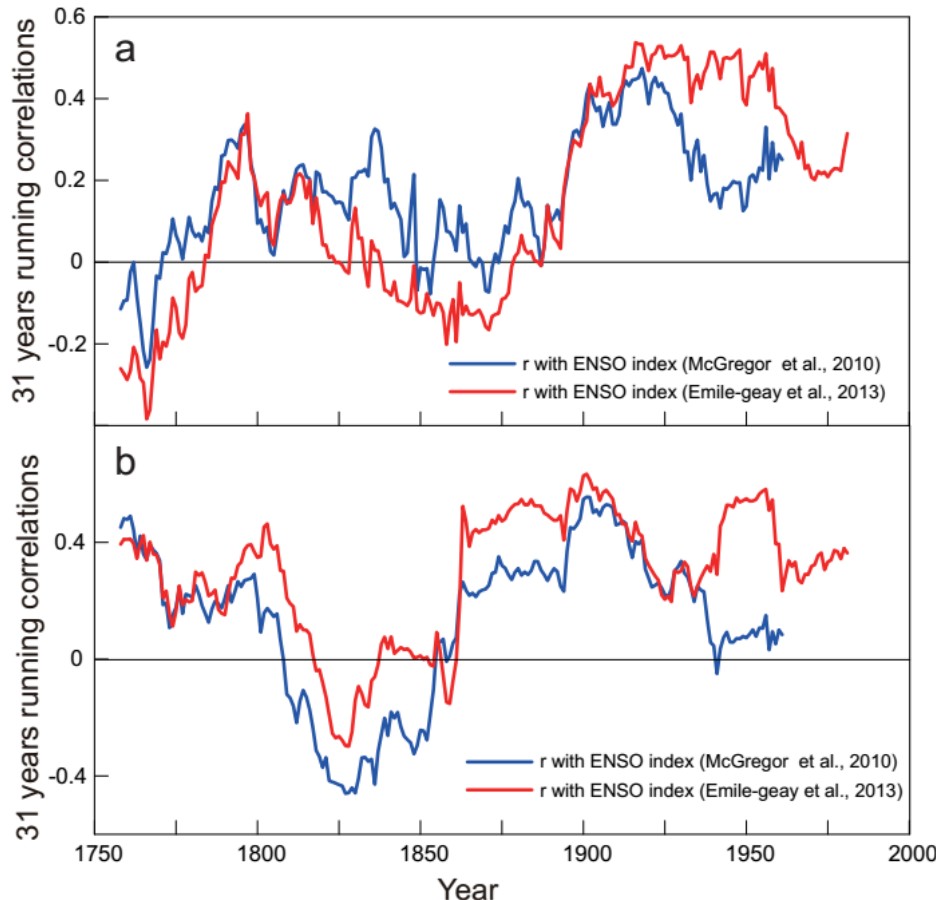

Figure 9. Thirty-one-year running correlations between two ENSO reconstructions and the H5 regional tree-ring $\delta^{18}O$ record

5    (a), and Indian Ocean SST reconstruction (b).

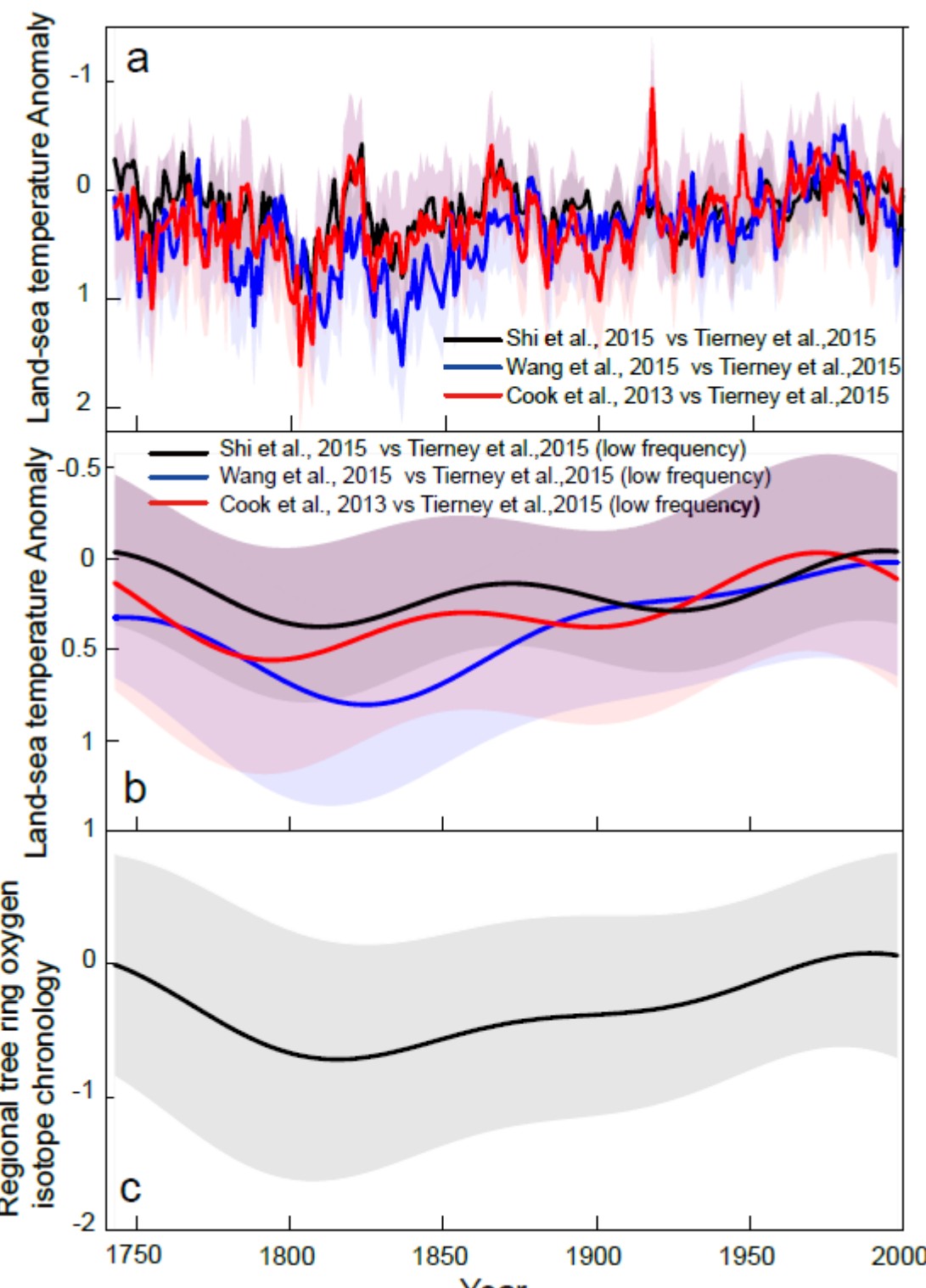

Figure 10. a: Land-ocean temperature anomaly based on three summer temperature reconstruction for the Tibetan Plateau and one Indian Ocean SST reconstruction; b and c: low frequency variations of land-sea thermal contrasts and the H5 regional tree-ring $\delta^{18}$O chronology. Shades with different colours indicate uncertainty for each time series.