# Peer review of "Decreasing Indian summer monsoon in northern Indian subcontinent during the last 180 years: evidence from five tree cellulose oxygen isotope chronologies"

_Climate of the Past, 2016_

## Referee Comment (RC3)

In their work, Xu et al. develop two new tree-rings isotopic chronologies of $\delta^{18}O$ from Northern India, and, combined with three other $\delta^{18}O$ chronologies, propose a multi-decadal regional reconstruction of the Indian summer monsoon. The regional record is further investigated using correlation and spectral analyses to document: 1) the drivers of Indian summer monsoon variability, and 2) the long-term trends of Indian summer monsoon intensity.

The new Data and this regional reconstruction are valuable to document a key hydroclimate component of the region, which lacks high resolution and long-term proxy records.

The methodology used in this paper as well as the results are robust. The data analyses, however, would benefit from further in depth exploration of each chronology signal. The discussion needs more regional scope but this can be improved if more data analyses are carried.

**General Comments**

- The authors present only correlations between the five $\delta^{18}O$ chronologies. How are the correlations for high and low frequency between the 5 chronologies?

- Climate correlations for each chronology $\delta^{18}O$ are summarized in the text however a figure of correlation between each chronology and the main climate factor (precipitation and/or PDSI) should be presented to assess the climate signal in each chronology before creating the composite regional signal.

- The composite signal (average of 5 centered $\delta^{18}O$ chronologies) should be presented with a standard deviation or an uncertainty term in order to see if the uncertainty changed over time. This will be important to discuss the relationship between the composite regional signal and its relation with Indian summer monsoon indices for the last ~ 200 years.

- The $\delta^{18}O$ signal is often described from a theoretical perspective. Could the authors provide more evidence of the $\delta^{18}O$ signal in these particular chronologies? Comparison with RH or source water isotopes? Or using process based evaluation by means of $\delta^{18}O$ forward modelling (Eg. Evans 2007).

- Analyses of observations would enhance the strength of the tree rings data: trends of rainfall for the region, as well as the various indices discussed in the paper. Additionally, $\delta^{18}O$ tree-rings and rainfall amount over the observations period should be plotted to strengthen the interpretation of the amount effect described in the results-discussion sections. This can be done for each site or at regional scale.

- In Page 9 paragraph 1 (~ line 5): Is the RH threshold 1%?

- In Page 9 paragraph 1 (~ line 5): further discussion is required when assessing how the source $\delta^{18}O$ is integrated differently between tree-rings and speleothem proxies.

- Page 10 from line 10 to 25: the text needs substantial editing, there are lot of repetitions and the discussion is not clear. Often the authors start describing the implication of their record and its comparison with regional records without an in-depth discussion.

- Page 9 starting line ~ 20. The discussion here is interesting, however, needs some clarification and editing.

For the clarification: 1) the authors report a decreasing $\delta^{18}O$ trend from 1743-1820. How many chronologies are included in this part of the record? According to Table 1 and Figure 4, only 2 chronologies extend back to 1743. Authors should use caution when making regional trend interpretations

2) an increasing $\delta^{18}O$ trend from 1820 to 2000 is observed in the $\delta^{18}O$ tree-rings interpreted as an increase in the Indian monsoon intensity, also observed from other regional proxies. Is this trend also observed when considering the chronologies individually? What are the statistics for this trend?

- Page 10, line 20. The discussion of factors (for instance aerosols) other than the decreasing land-ocean thermal contrast and their role in the decreasing Indian Monsoon intensity needs to be more detailed. What is the land-ocean thermal contrast resolution, interannual? Decadal? From Figure 11 the proxy records for ocean and land temperature do not seem to have the same temporal resolution.

- when assessing the ENSO-and H5 regional $\delta^{18}O$ correlations, a 31-year window is too large (ENSO is 2-7 years). It would be helpful to investigate ENSO- and Tree-rings $\delta^{18}O$ correlation for individual chronologies to test whether the decorrelation is observed for all chronologies which reflect sites under slightly different precipitation regime and Indian summer monsoon influence (based on Fig 1).

**Specific comments**

- In Results and Discussion, section 3.1. The standard deviation of individual tree-ring cores can be added next to the mean.

- In Table 1 the mean $\delta^{18}O$ for each chronology should have a unit (‰)

- In Fig 1 and Table 2 the chronologies from previous studies have a different name. Bhutan in -Table 2 and Wache in Fig 1.

- In Fig 2 the triangle and circle symbols have no legend.

- In Fig 3 the triangle and circle symbols legend is too big. This can be reduced to only the symbols without the line (same should be applied to Fig 2).

- In Fig 4, the sample depth (number) should be added in a graph below the composite time-series since the number of averaged trees over time is not the same (prior to 1740 and after 2000).

- In Fig 5, add the location of all the sites to help visualize the strength of the field correlations.

- In Fig 8, add the location of the 18O chronologies for spatial reference of the SST correlations.

-Page 8 line 1, a word or punctuation is missing after the parenthesis (eastern-Pacific el Nino).

---

## Referee Comment (RC1) · Anonymous Referee #1 · 4 Mar 2017

The authors attempted to reconstruct long-term Indian monsoon rainfall or intensity variations based on tree-ring oxygen isotope chronologies produced from northern part of the Indian subcontinent. They developed two new isotope chronologies spanning the past two centuries or more. Combined with existing tree-ring oxygen isotope records, they produced a so-called regional composite chronology which is regarded as an indicator of the Indian summer monsoon intensity. Further, the authors explored variation characteristics of the regional composite and potential driving mechanism by the way of statistical comparison. One main conclusion is that the intensity of the Indian summer monsoon exhibits a decreasing trend since the early nineteenth century. However,

this conclusion is difficult to assess, and it might be subject to large uncertainties since they did not show the comparison of all available five tree-ring oxygen records. Furthermore, observed all Indian rainfall record does not show a decreasing trend according to much longer observational data. It needs much more evidence to validate. At any rate, given that the high-resolution proxy records are still sparse in this region, their effort is expected to add new contribution to the knowledge of regional climate variability. In general the methodology used here is simple and routine in dendrochronological study. This work is worthy of publication in the Climate of the Past. However, there are some questions should be clarified or explained before it is ready to go.

Specific comments:

1) The main conclusion in this manuscript is that the Indian summer monsoon strength decreased during the last 180 years. To confirm the robustness of this conclusion, much more evidence and discussion are needed to compliment using all available proxy records together with some statistical approach. Furthermore, it is necessary to show a comparison of five isotope chronologies from all five sites. In so soing, one can see how different they are on low-frequency variations, particularly for the time span from 1820 AD till at present. Another concern is why the authors do not use any ring width chronologies into discussion since a large number of chronologies have been published in the study region. It is no problem for ring width to retain century-scale climatic signals due to long-lived species used in producing the chronologies. 2) The overall length of the record is approximately 300 yr or so, so it is impossible to locate a cycle of 350 yr in the composite record. 3) ENSO has the strongest power in wave length or cycles 2-7 yr, so 31-yr moving correlations are not suitable in this case. 4) Shi et al. (2015) temperature reconstruction represents a 10-year moving average rather than a yearly time resolution. It is suggested that the authors also use other summer season temperature reconstructions (see Cook et al., 2013, CD; Wang et al., 2015, JC). In so doing, it can be regarded as a sensitivity experiment. 5) It is strange why the best spatial correlations between the regional tree ring $\delta$18O record with May-

September SST during 1871-2008 CE (Fig. 8) center around the central tropical ocean rather than in the eastern tropical ocean due to a close association between the ENSO and the Indian monsoon. 6) Fig. 10 is not helpful due to poor agreement between the two curves.

———————————————

---

## Short Comment (SC1) · 7 Mar 2017

This comment was prepared through a group discussion of the SPATIAL laboratory at the University of Utah.

Overview: Xu et al. provide a new stacked record of tree ring cellulose oxygen isotopes from five locations along the southern Himalaya, spanning a time range of 1743-2008 CE. They find significant correlations with regional climate indices of precipitation and Indian monsoon strength over the instrumental record, and from this, infer that their stacked record can be used to reconstruct the strength of the Indian monsoon prior to

the industrial record. From this, they draw two potentially exciting conclusions from their analysis: (a) high-frequency ENSO variability (e.g., periods of 2.4-5 years in their figure 7) may be recorded in the stacked tree ring dataset and (b) low-frequency centennial scale variability (e.g., periods of 160-350 years in their figure 7) may reflect long-term variability in monsoon strength, which they support by an analysis of long-term changes in the land-sea temperature contrast derived from proxy records. However, the authors do not provide information on uncertainty and error, and therefore, it is difficult to assess the robustness of their conclusions. Our view is that this data merits publication, but that considerable revisions need to be made to help clarify their analyses and support their conclusions. Therefore, we recommend acceptance pending major revisions described below.

Major comments: (1) Error and uncertainty are not adequately explored or explained. We provide several examples of analyses in the paper that would benefit from a more thorough treatment of error and uncertainty propagation:

No uncertainty is given on the individual chronologies provided (e.g., the uncertainty from combining individual trees at a location to estimate the average delta18O record at that location), nor is it propagated to the averaged delta18O chronology of the stacked record in figure 4.

The authors make several comparisons between their stacked tree ring record and other proxy indicators of ENSO (Fig. 9), stalagmite oxygen isotopes (Fig 10), and Indian Ocean SSTs and Tibetan Plateau temperatures (Fig 11). However, they do not consider either the potential errors in proxy reconstructed values (e.g., the error in reconstructed SSTs), nor potential errors in the age model used to assign a date to those proxy values. As a result, it is difficult to assess how robust the signals they derive from comparisons between proxy records are, and how they compare to the variability. For example, in figure 11, it is not clear that the reconstructed land-sea temperature anomaly is a substantial, robust, or significant deviation from zero if estimates of uncertainty are absent.

(2) The rationale for why the authors think that their stacked record reflects regional changes in the monsoon is absent – the signals from each location appear coherent, but it is not shown that they are actually coherent. There's a wide range in correlation coefficients between sites in Table 2, where the lowest correlation coefficients suggest that sampling at Manali only explains ∼5% of the variance observed at Bhutan. Thus, while we find the possibility that these sites record a regional-scale signal to be exciting, the rationale for combining all of these datasets for a regional interpretation should be explained in more detail.

Additionally, the authors hint that the relationship between sites may not be stationary (pg 5, L18-21), as they note decadal-scale changes are often not observed coherently through their stacked record. The analysis would benefit from exploring potential reasons for why this might be the case – are there other potential explanations than variations in the ISM?

(3) The authors draw conclusions that may not be supported by their time series analysis methods. A section describing the spectral analysis methods, software, etc. that were used should be added to the methods section so that their results could be replicated by other researchers. Additionally, it is not clear how the authors determined the significance levels plotted in Figure 7 – this should be explained. A few additional comments/questions regarding the time series analysis are provided below:

The conclusion that their record captures centennial-scale variability requires more justification considering their record is only 350 years long. They claim significant spectral power at 160 and 350 year intervals (Fig 7) – though the 350 year peak is the secular trend in their 350 year dataset, and the 160 year cycle may also not be significant – more details about how significance levels are determined should be provided.

It was not clear why a 31-year moving correlation was used in figure 9 – could you expand on this choice?

(4) Writing is imprecise and organization needs improvement. We have provided a few

of the more pressing examples to help guide revisions:

- The methods section requires substantial additions. First, the time series analysis methods used need to be described in an additional subsection, and in enough detail that other researchers could recreate the analysis. Second, many of the paragraphs in the results start with a description of how an analysis was done – these should be moved to the methods section.

- The introduction brings up several relevant factors about the ISM without relating them directly to this study. This section would be more impactful if it were better focused on what is known about how these factors influence the ISM rather than just providing a list. This addition would help clarify how your results improve our understanding of the ISM.

- The discussion section analyzing why there may be a weakening monsoonal circulation over last few hundred years requires a more in-depth analysis. The presented tree ring records cannot answer this question, and the question diverges from the main focus of the paper. The land-sea contrast mechanism described is potentially interesting, but the authors need to be more descriptive about: (a) how well do we know there has been a change in the land-sea temperature contrast? (b) are there other potential explanations, given the long list of factors influencing the ISM the authors list in the introduction?

Following the SPATIAL laboratory group discussion, Rich Fiorella compiled this short comment based on input from Gabe Bowen, Rose Smith, Annie Putman, Crystal Tulley-Cordova, Chao Ma, Zhongyin Cai, Yusuf Jameel, Brenden Fischer-Femal, and Sagarika Banerjee.

————————————————

---

## Referee Comment (RC2) · Anonymous Referee #2 · 22 Mar 2017

In their work, combining a new tree-ring isotope record with the existing records, Chenxi et al have constructed a regional tree ring cellulose oxygen isotope record for the northern Indian Subcontinent. The authors further show correlation between the tree-ring isotopic record and various indices of the monsoonal strength. After establishing this coherence, the authors further use the tree-ring isotope record for understanding long-term variation in monsoonal precipitation. Overall the manuscript is written well and arguments are coherently presented. I recommend publishing the manuscript with minor revision.

Following points should be considered while revising the manuscript.

Section 3.3 is too long to read. Consider subdividing into smaller sections.

Page 9 first paragraph : epikarst dynamics could be more responsible for the incoherence of the two records. Some discussion is required.

It would be helpful for the reader if authors describe the nature of long-term variations in modern instrumental rainfall data. Analysis by Sontakke et al Holocene 2008 and Bhutiyani et al IJC 2010 could be helpful. In fact, the latter article also points out to a significant decreasing trend since 1866 in the monsoonal rainfall.

The way regional isotope record is constructed (average of averages of d18O records of different sites) underestimates the uncorrelated variability. Quantification regarding this should be added to Table 2.

---

## Author Comment (AC1) · 18 Apr 2017

**Referee #1**

The authors attempted to reconstruct long-term Indian monsoon rainfall or intensity variations based on tree-ring oxygen isotope chronologies produced from northern part of the Indian subcontinent. They developed two new isotope chronologies spanning the past two centuries or more. Combined with existing tree-ring oxygen isotope records, they produced a so-called regional composite chronology which is regarded as an indicator of the Indian summer monsoon intensity. Further, the authors explored variation characteristics of the regional composite and potential driving mechanism by the way of statistical comparison. One main conclusion is that the intensity of the Indian summer monsoon exhibits a decreasing trend since the early nineteenth century. However, this conclusion is difficult to assess, and it might be subject to large uncertainties since they did not show the comparison of all available five tree-ring oxygen records. Furthermore, observed all Indian rainfall record does not show a decreasing trend according to much longer observational data. It needs much more evidence to validate. At any rate, given that the high-resolution proxy records are still sparse in this region, their effort is expected to add new contribution to the knowledge of regional climate variability. In general the methodology used here is simple and routine in dendrochronological study. This work is worthy of publication in the Climate of the Past. However, there are some questions should be clarified or explained before it is ready to go.

Specific comments:
1) The main conclusion in this manuscript is that the Indian summer monsoon strength decreased during the last 180 years. To confirm the robustness of this conclusion, much more evidence and discussion are needed to compliment using all available proxy records together with some statistical approach. Furthermore, it is necessary to show a comparison of five isotope chronologies from all five sites. In so soing, one can see how different they are on low-frequency variations, particularly for the time span from 1820 AD till at present. Another concern is why the authors do not use any ring width chronologies into discussion since a large number of chronologies have been published in the study region. It is no problem for ring width to retain century scale climatic signals due to long-lived species used in producing the chronologies.

**Answer**: In previous manuscript, we compared five isotope chronologies from all five sites in Figure 4a. Based on your helpful suggestion, we added the comparisons among five oxygen isotope chronologies from all sites in new Figure 4. Please see the new Figure 4 in revised manuscript or the following Figure (red line: 31-year running average; black line: mean value

oxygen isotopes from all trees; gray shadows: the 95% (±1.96σ) confidence limits to show the uncertainty of each chronology and regional tree ring oxygen isotope chronology, respectively (except for tree ring oxygen isotope chronology in Hulma, western Nepal, because the tree ring oxygen isotopes chronology was produced by pooling method, and therefore the uncertainty of this chronology was not able to show.).

[Figure]

Figure Caption: Figure 4: Tree ring oxygen isotope chronologies from five sites (a-e) and the regional tree ring oxygen isotope chronology (f). (black line: mean values for all samples; red line: 31-year running average for the chronology; gray shadows: the 95% (±1.96σ) confidence limits)

Yes, there are many ring width chronologies in the study area. There are two reasons that we do not use these ring width chronologies for reconstructing Indian summer monsoon history. 1) Most of ring width chronologies in the study area mainly reflected the climate in spring rather than summer. For example, Cook et al., (2003, International Journal of climatology) reconstructed Kathmandu Temperature based on ring width chronologies. Although Yadav et al., (2014, Quaternary International) reconstructed Februarye-May precipitation using tree-ring width data of Himalayan cedar. Trees throughout the region do not show any direct responses to summer monsoon rainfall (June–September). 2) Teak from Kerala, Southern India reveals that low growth years (narrow rings) are significantly associated with deficient Indian rainfall (DIR). However, normal or above normal rainfall is not consistently reflected as higher tree growth, possibly due to a moisture threshold being reached, above which trees can no longer respond (Borgaonkar et al., 2010, Paleo3). Because the higher tree growth cannot record the signal of strong rainfall, ring width chronology of teak in Southern India was not used for Indian summer monsoon reconstruction.

Borgaonkar H, Sikder A, Ram S, et al. El Nino and related monsoon drought signals in 523-year-long ring width records of teak (Tectona grandis LF) trees from south India. Palaeogeography Palaeoclimatology Palaeoecology, 2010, 285(1):74-84.
Cook E R, Krusic P J, Jones P D. Dendroclimatic signals in long tree-ring chronologies from the Himalayas of Nepal. International Journal of Climatology, 2003, 23(7):707-732.
Yadav R R, Misra K G, Kotlia B S, et al. Premonsoon precipitation variability in Kumaon Himalaya, India over a perspective of ~300 years. Quaternary International, 2014, 325:213-219.

2)  The overall length of the record is approximately 300 yr or so, so it is impossible to locate a cycle of 350 yr in the composite record.

**Answer**: Thanks for your helpful comments. We checked the codes that were used to calculate the Power Spectra based on the multi-taper method in previous manuscript. The confidence level in low frequency have some problems. We recalculate power spectra based on the multi-taper method using the Software "kSpectra Toolkit"(v3.4). The results show that the H5 regional tree ring $\delta^{18}O$ record contains several high-frequency periodicities (4 and 5 years), as well as lower frequency periodicities (~133 years). Please see new Figure 7 in revised manuscript or the following Figure.

[Figure]

Figure Caption: Figure 7: Multi-taper power spectra for the H5 regional tree ring $\delta^{18}O$ record.

*3.3 Interannual variability of the ISM inferred from the regional tree ring $\delta^{18}O$ record*

The results of spectral analysis using the multi-taper method (Mann and Lees, 1996) indicates that the H5 regional tree ring $\delta^{18}O$ record contains several high-frequency quasi-periodicities (4 and 5 years), as well as lower frequency periodicities (~133 years) at a confidence level greater than 99% (Figure 7).

3) ENSO has the strongest power in wave length or cycles 2-7 yr, so 31-yr moving correlations are not suitable in this case.

**Answer**: Maybe our explanation on 31-year moving correlation is not so clear. ENSO-Monsoon teleconnection is known to be a nonstationary process. We need to evaluate how the

relationships between ENSO and monsoon changed in the past. Because ENSO has the strongest power in wave length or cycles 2-7 yr, 31-year or 21-year window moving correlations between ENSO and precipitation were used to evaluate the stability of relationship between ENSO and climate. The 31-year and 21-year window can cover the main cycles (2-7 years) of ENSO. For example, Camberlin et al., (2004, Climate Dynamics) used 31-year moving correlations between NINO3 SST and seasonal rainfall anomalies over a few regions to see ENSO/rainfall teleconnections. 21-year moving window correlation analysis between ENSO and climate was used to investigate the relationship between ENSO and East Asian winter monsoon/precipitation and flood pulse in the Mekong River Basin (Kim et al., 2016; Räsänen and Kummu, 2013)

Camberlin P, Chauvin F, Douville H, et al. Simulated ENSO-tropical rainfall teleconnections in present-day and under enhanced greenhouse gases conditions. Climate Dynamics, 2004, 23(6):641-657.

Kim J W, An S I, Jun S Y, et al. ENSO and East Asian winter monsoon relationship modulation associated with the anomalous northwest Pacific anticyclone. Climate Dynamics, 2016:1-23.

Räsänen T A, Kummu M. Spatiotemporal influences of ENSO on precipitation and flood pulse in the Mekong River Basin. Journal of Hydrology, 2013, 476(1):154-168.

4) Shi et al. (2015) temperature reconstruction represents a 10-year moving average rather than a yearly time resolution. It is suggested that the authors also use other summer season temperature reconstructions (see Cook et al., 2013, CD; Wang et al., 2015, JC). In so doing, it can be regarded as a sensitivity experiment.

**Answer**: Thanks for your suggestions. We added two summer temperature reconstructions (Cook et al., 2013 and Wang et al., 2015) to calculate the land-sea thermal contrasts (New Figure 10 in revised manuscript). Three land-see temperature anomaly time series (Land temperature: Cook et al., 2013; Shi et al., 2015; Wang et al., 2015; Sea temperature: Tierney et al., 2015) showed similar lower frequency variations. A decreasing trend of land-sea temperature anomaly during the last 200 years were shown by three land-see temperature anomaly time series. In addition, our regional tree ring $\delta^{18}O$ chronology showed an increasing trend (reduced intensity of Indian summer monsoon) since 1820 CE.

[Figure]

Figure Caption: Figure 10. a: Land-sea Temperature Anomaly based on three summer temperature reconstruction for the Tibetan Plateau and one Indian Ocean SST reconstruction; b and c: centennial variations of land-sea thermal contrasts and the H5 regional tree ring $\delta^{18}O$ chronology.

5) It is strange why the best spatial correlations between the regional tree ring _18O record with May-September SST during 1871-2008 CE (Fig. 8) center around the central tropical ocean rather than in the eastern tropical ocean due to a close association between the ENSO and the Indian monsoon.

**Answer**: Historical rainfall record and atmospheric general circulation model experiments showed that El Nino events with the warmest sea surface temperature (SST) anomalies in the central equatorial Pacific are more effective in weakening the Indian monsoon severely than events with the warmest SSTs in the eastern equatorial Pacific (Kumar et al., 2006, Science). Kumar K K, Rajagopalan B, Hoerling M, et al. Unraveling the mystery of Indian monsoon failure during El Nino. Science, 2006, 314(5796):115-119.

6) Fig. 10 is not helpful due to poor agreement between the two curves.

**Answer**: We have added uncertainty of age for stalagmite samples and more discussion on the similarities and dissimilarities between regional tree ring oxygen isotope chronology and stalagmite oxygen isotope time series in northern India. Please see the revised Figure 11 and Section 3.5 in the revised manuscript.

*3.5 Comparison of regional tree ring $\delta^{18}O$ record with speleothem $\delta^{18}O$ record in northern India*

The H5 regional tree ring $\delta^{18}O$ record does not exhibit significant decadal to multi-decadal periodicities (Figure 7), while the main spectral component of high-resolution speleothem $\delta^{18}O$ records (a proxy of ISM rainfall in northern and central India) consists of multi-decadal periodicities (~15, 20, 30, 60 and 70 years) (Sinha et al., 2011; Sinha et al., 2015). This inconsistency may be the result of the different types of proxy record used together with micro-environmental differences between the sampling sites. Although decadal to multi-decadal variability of the H5 tree ring $\delta^{18}O$ record is not strongly developed, the record does contain decadal to multi-decadal changes. Decadal to multi-decadal variability was extracted using bandpass filters (15-80 years) (Figure 11, red line). From the perspective of decadal to multi-decadal changes, the H5 record shares similarities with the speleothem record, while the H5 record are out-of-phase with speleothem $\delta^{18}O$ records during several intervals (Figure 11).

Based on the oxygen isotope fractionation theory, tree ring $\delta^{18}O$ and speleothem $\delta^{18}O$ should share similar changes (Managave, 2014) if both of them inherit a common source water $\delta^{18}O$ signal, as shown by Ramesh, et al (2013). The following reasons may cause incoherence between regional tree ring $\delta^{18}O$ and speleothem $\delta^{18}O$. Other controlling factors differentially affect tree ring $\delta^{18}O$ and speleothem $\delta^{18}O$ values. Relative humidity has an important impact on tree ring $\delta^{18}O$ in regions where the variation of relative humidity during the growing season exceeds 1% (Managave, 2014), while the cave epikarst dynamics affect speleothems $\delta^{18}O$ significantly (Lachniet, 2009). The infiltrating water from different rainfall events may be stored and mixed in the epikarst. Lag times of $\delta^{18}O$ values in drip waters relative to rainfall are several years or decades in some locations (Lachniet, 2009), and a slow transit time smoothed climate signal. In addition, limited three [230]Th dates points (3 control points) and relative large age uncertainty (9-31 years) of speleothems $\delta^{18}O$ time series during the common period of 1743-2000 may result in the incoherence between tree ring and speleothems $\delta^{18}O$. Long-term process-based study on tree ring $\delta^{18}O$ and speleothem $\delta^{18}O$ variations in future study are needed for a better understanding for climatic implication of two proxies.

---

## Author Comment (AC2) · 18 Apr 2017

**Referee #2**

In their work, combining a new tree-ring isotope record with the existing records, Chenxi et al have constructed a regional tree ring cellulose oxygen isotope record for the northern Indian Subcontinent. The authors further show correlation between the tree-ring isotopic record and various indices of the monsoonal strength. After establishing this coherence, the authors further use the tree-ring isotope record for understanding long term variation in monsoonal precipitation. Overall the manuscript is written well and arguments are coherently presented. I recommend publishing the manuscript with minor revision.

Following points should be considered while revising the manuscript.
(1) Section 3.3 is too long to read. Consider subdividing into smaller sections.

**Answer**: Thanks for your helpful suggestions. The previous Section 3.3 was divided into three parts. New Section 3.3 (Interannual variability of the ISM inferred from the regional tree ring $\delta^{18}O$ record), Section 3.4 (Centennial variability of the ISM inferred from the regional tree ring $\delta^{18}O$ record) and Section 3.5 (Comparison of regional tree ring $\delta^{18}O$ record with speleothem $\delta^{18}O$ record in northern India).

(2) Page 9 first paragraph: epikarst dynamics could be more responsible for the incoherence of the two records. Some discussion is required.

**Answer**: Thanks for your helpful suggestions. We added the related discussion on epikarst dynamics. Please see the following paragraph.

*3.5 Comparison of regional tree ring $\delta^{18}O$ record with speleothem $\delta^{18}O$ record in northern India*

The H5 regional tree ring $\delta^{18}O$ record does not exhibit significant decadal to multi-decadal periodicities (Figure 7), while the main spectral component of high-resolution speleothem $\delta^{18}O$ records (a proxy of ISM rainfall in northern and central India) consists of multi-decadal periodicities (~15, 20, 30, 60 and 70 years) (Sinha et al., 2011; Sinha et al., 2015). This inconsistency may be the result of the different types of proxy record used together with micro-environmental differences between the sampling sites. Although

decadal to multi-decadal variability of the H5 tree ring $\delta^{18}$O record is not strongly developed, the record does contain decadal to multi-decadal changes. Decadal to multi-decadal variability was extracted using bandpass filters (15-80 years) (Figure 11, red line). From the perspective of decadal to multi-decadal changes, the H5 record shares similarities with the speleothem record, while the H5 record are out-of-phase with speleothem $\delta^{18}$O records during several intervals (Figure 11).

Based on the oxygen isotope fractionation theory, tree ring $\delta^{18}$O and speleothem $\delta^{18}$O should share similar changes (Managave, 2014) if both of them inherit a common source water $\delta^{18}$O signal, as shown by Ramesh, et al (2013). The following reasons may cause incoherence between regional tree ring $\delta^{18}$O and speleothem $\delta^{18}$O. Other controlling factors differentially affect tree ring $\delta^{18}$O and speleothem $\delta^{18}$O values. Relative humidity has an important impact on tree ring $\delta^{18}$O in regions where the variation of relative humidity during the growing season exceeds 1% (Managave, 2014), while the cave epikarst dynamics affect speleothems $\delta^{18}$O significantly (Lachniet, 2009). The infiltrating water from different rainfall events may be stored and mixed in the epikarst. Lag times of $\delta^{18}$O values in drip waters relative to rainfall are several years or decades in some locations (Lachniet, 2009), and a slow transit time smoothed climate signal. In addition, limited three [230]Th dates points (3 control points) and relative large age uncertainty (9-31 years) of speleothems $\delta^{18}$O time series during the common period of 1743-2000 may result in the incoherence between tree ring and speleothems $\delta^{18}$O. Long-term process-based study on tree ring $\delta^{18}$O and speleothem $\delta^{18}$O variations in future study are needed for a better understanding for climatic implication of two proxies.

(3) It would be helpful for the reader if authors describe the nature of long-term variations in modern instrumental rainfall data. Analysis by Sontakke et al Holocene 2008 and Bhutiyani et al IJC 2010 could be helpful. In fact, the latter article also points out to a significant decreasing trend since 1866 in the monsoonal rainfall.

**Answer**: Thanks for your helpful suggestions. We have added the description of nature of long-term variations in modern instrumental rainfall data based on the Sontakke et al., (2008) and Bhutiyani et al., (2010) in the Introduction and Section 3.4 (*Centennial*

[revised manuscript text omitted]

(4) The way regional isotope record is constructed (average of averages of d18O records of different sites) underestimates the uncorrelated variability. Quantification regarding this should be added to Table 2.

**Answer**: Thanks for your helpful suggestions. We have added the uncertainty in regional tree ring oxygen isotope chronology in Figure 4f to evaluate the inter-site variability. In addition, we checked the uncorrelated variability by comparison between regional tree ring oxygen isotope chronology and PC1 of five tree ring oxygen isotope chronologies in northern Indian sub-continent. The regional tree ring oxygen isotope chronology is highly correlated with PC1 of five tree ring oxygen chronologies (r=0.998, n=200, p<0.001), which indicates that regional tree ring oxygen isotope chronology reflect the main common signal of five tree ring oxygen isotope chronologies.

---

## Author Comment (AC3) · 18 Apr 2017

**Short Comments:**

This comment was prepared through a group discussion of the SPATIAL laboratory at the University of Utah.

Overview: Xu et al. provide a new stacked record of tree ring cellulose oxygen isotopes from five locations along the southern Himalaya, spanning a time range of 1743-2008 CE. They find significant correlations with regional climate indices of precipitation and Indian monsoon strength over the instrumental record, and from this, infer that their stacked record can be used to reconstruct the strength of the Indian monsoon prior to the industrial record. From this, they draw two potentially exciting conclusions from their analysis: (a) high-frequency ENSO variability (e.g., periods of 2.4-5 years in their figure 7) may be recorded in the stacked tree ring dataset and (b) low-frequency centennial scale variability (e.g., periods of 160-350 years in their figure 7) may reflect long-term variability in monsoon strength, which they support by an analysis of long-term changes in the land-sea temperature contrast derived from proxy records. However, the authors do not provide information on uncertainty and error, and therefore, it is difficult to assess the robustness of their conclusions. Our view is that this data merits publication, but that considerable revisions need to be made to help clarify their analyses and support their conclusions. Therefore, we recommend acceptance pending major revisions described below.

Major comments:

(1) Error and uncertainty are not adequately explored or explained.

We provide several examples of analyses in the paper that would benefit from a more thorough treatment of error and uncertainty propagation: No uncertainty is given on the individual chronologies provided (e.g., the uncertainty from combining individual trees at a location to estimate the average delta18O record at that location), nor is it propagated to the averaged delta18O chronology of the stacked record in figure 4. The authors make several comparisons between their stacked tree ring record and other proxy indicators of ENSO (Fig. 9), stalagmite oxygen isotopes (Fig 10), and Indian Ocean SSTs and Tibetan Plateau temperatures (Fig 11). However, they do not consider either the potential errors in proxy reconstructed values (e.g., the error in reconstructed SSTs), nor potential errors in the age model used to assign a date to those proxy values. As a result, it is difficult to assess how robust the signals they derive from comparisons between proxy records are, and how they compare to the variability. For example, in figure 11, it is not clear that the reconstructed land-sea temperature anomaly is a substantial, robust, or significant deviation from zero if estimates of uncertainty are absent.

**Answer**: Thanks for your suggestions. We have added the 95% ($\pm1.96\sigma$) confidence limits of different tree ring oxygen isotope time series in Manali, JG, Ganesh and Wache as the uncertainty of inter-tree variability, which are shown in New Figure 4a,b,d,e in revised manuscript by gray shadows. For tree ring oxygen isotopes data in Hulma, we cannot evaluate the inter-tree oxygen isotope variability, because tree ring oxygen isotope chronology in Hulma was built up by pooling method. The uncertainty of regional tree ring oxygen isotope chronology was evaluated by showing the 95% ($\pm1.96\sigma$) confidence limits (Figure 4f). Please see the revised Figure 4.

[Figure]

Figure Caption: Figure 4: Tree ring oxygen isotope chronologies from five sites (a-e) and the regional tree ring oxygen isotope chronology (f). (black line: mean values for all samples; red line: 31-year running average for the chronology; gray shadows: the 95% (±1.96σ) confidence limits)

For the uncertainty of age model for stalagmite oxygen isotope time series, we have added the dating results and uncertainty of stalagmite oxygen isotope data in northern India in new Figure 11. During the common period (1743-2000) between regional tree ring oxygen isotope chronology and stalagmite oxygen isotope data, there are three dating point with uncertainty in range of 9~31 years.

[Figure]

New Figure 11. Comparison between multi-decadal regional tree ring $\delta^{18}O$ variations (red line) with stalagmite $\delta^{18}O$ changes (black line) in northern India. Rhombus with error indicates the $^{230}$Th dates with uncertainty in stalagmite $\delta^{18}O$ chronology.

To check robustness of the low-frequency land-sea temperature anomaly, three different temperature reconstruction (Cook et al., 2013; Shi et al., 2015; Wang et al., 2015) in Tibetan Plateau and one Indian Ocean SST reconstruction was used. Three land-see temperature anomaly time series showed similar lower frequency variations. A decreasing trend of land-sea temperature anomaly during the last 200 years were shown by three land-see temperature anomaly time series. In addition, we added the ± 1 RMSE (root mean square error) as

uncertainties of each temperature reconstruction in Tibetan Plateau and Indian Ocean. The uncertainty of land-see temperature anomaly was calculating by adding RMSE from land and sea temperature reconstruction, which was shown by shadows in New Figure 10 in the revised manuscript.

[Figure]

Figure Caption: Figure 10. a: Land-sea Temperature Anomaly based on three summer temperature reconstruction for the Tibetan Plateau and one Indian Ocean SST reconstruction;

b and c: centennial variations of land-sea thermal contrasts and the H5 regional tree ring $\delta^{18}O$ chronology.

(At the end of Section 3.5 of the revised manuscript)

Several studies show that increased Indian Ocean SSTs caused a reduction in ISM rainfall (Fan et al., 2009; Naidu et al., 2009; Sun et al., 2016). The Indian Ocean SST has increased since 1840-1860 CE (Tierney et al., 2015; Wilson et al., 2006), which supports this explanation. Although the SST of the Indian Ocean significantly affects the ISM, the land-sea thermal contrast is also an important influencing factor (Roxy et al., 2015). In particular, heating anomalies over the Tibetan Plateau have a significant influence on the ISM via their effect on the atmospheric temperature gradient between the Tibetan Plateau and the tropical Indian Ocean (Fu and Fletcher, 1985; Sun et al., 2010). The history of land-sea thermal contrasts is reconstructed based on temperature differences between the Tibetan Plateau and the Indian Ocean (Figure 10a), and centennial variations in this record are shown in Figure 10b. Three reconstructed land-sea thermal contrasts showed a decreasing trend since 1800 CE and 1820 CE (Figure 10b), and the H5 record exhibits a similar pattern of changes on a centennial scale (Figure 10c). The decreasing land-sea thermal contrast since 1800 and 1820 CE has resulted in a weaker ISM, and the increasing trend of the H5 record since 1820 CE also indicates a reduced ISM intensity. In addition, aerosol emissions may be another reason to cause weakened ISM. Because, aerosol emissions could result in a slowdown of the tropical meridional overturning circulation, cooler temperatures over Europe and Asia relative to the ambient oceans, and a corresponding weakening of the ISM circulation (Bollasina et al., 2011; Cowan and Cai, 2011).

(2) The rationale for why the authors think that their stacked record reflects regional changes in the monsoon is absent – the signals from each location appear coherent, but it is not shown that they are actually coherent. There's a wide range in correlation coefficients between sites in Table 2, where the lowest correlation coefficients suggest that sampling at Manali only explains _5% of the variance observed at Bhutan. Thus, while we find the possibility that these sites record a regional-scale signal to be exciting, the rationale for combining all of these datasets for a regional interpretation should be explained in more detail. Additionally, the authors hint that the relationship between sites may not be stationary (pg 5, L18-21), as they note decadal-scale changes are often not observed coherently through their stacked record. The analysis would benefit from exploring potential reasons for why this might be the case - are there other potential explanations than

variations in the ISM?

**Answer**: The rationale for combing five tree ring oxygen isotope chronologies in monsoon area is that: tree ring oxygen isotopes in five sites show significant correlations with summer precipitation/PDSI, and summer climate in five sampling areas are controlled by Indian summer monsoon, so combining five oxygen isotope chronologies should be helpful to obtain monsoon-related information. The significant correlations between regional tree ring oxygen isotope chronology and all Indian monsoon/Indian summer monsoon Index/grid summer precipitation indicated that regional tree ring oxygen isotope chronology can reflect Indian summer monsoon changes. Given long distances (around 1400 km) between Manali and Buthan, correlation coefficient (r=0.23, p<0.001) is not bad. On the decadal-scale changes between different tree ring oxygen isotope chronologies, we discussed on it in another paper (Sano et al., under review). In addition, this is not the main topic of this paper.

We added the paragraph on the rationale for combing five tree ring oxygen isotope chronologies in monsoon area. Please see the following paragraph.

*3.1 Tree ring $\delta^{18}O$ variations in the southern Himalaya and a regional tree ring $\delta^{18}O$ record*

The oxygen isotopes of four individuals of *Abies spectabilis* in Ganesh (GE, central Nepal) and three individuals of *Cedrus deodara* in Jageshwar (JG, northern India) were measured for the interval from 1801-2000 CE and 1643-2008 CE, respectively. Individual tree ring $\delta^{18}O$ time series from four cores from central Nepal are shown in Figure 2a. The mean values of the $\delta^{18}O$ time series from 224c, 233b, 235b, and 226a are 23.09‰, 22.66‰, 21.87‰, and 22.94‰, respectively, from 1901-2000 CE; the standard deviations are 1.22‰, 1.27‰, 1.12‰, and 1.42‰, respectively. The inter-tree differences in $\delta^{18}O$ values are small. The $\delta^{18}O$ values of the four cores exhibit peaks in 1813. The mean inter-series correlations (Rbar) among the cores range from 0.56-0.78 (Figure 2c), based on a 50-year window over the interval from 1801-2000 CE.

Three tree ring $\delta^{18}O$ time series from northern India (JG) are shown in Figure 3a. The mean values of the $\delta^{18}O$ time series from 101c, 102c, and 103a are 30.11‰, 29.7‰ and 29.47‰, respectively, over the interval from 1694-2008 CE; the standard deviations are 1.49‰, 1.62‰ and 1.53‰, respectively. Three tree ring $\delta^{18}O$ time series in JG exhibit a consistent pattern of variations. The mean inter-series correlations (Rbar) among the cores range from 0.61-0.78

(Figure 3c), based on a 50-year window over the interval from 1641-2008 CE.

In northern Indian sub-continent, three long-term tree ring $\delta^{18}O$ chronologies from northwest India, eastern Nepal and Bhutan have been built up in previous studies (Sano et al., 2011; Sano et al., 2013; Sano et al., submitted, Figure 4). Two tree ring $\delta^{18}O$ chronologies in this study and three tree ring $\delta^{18}O$ chronologies in previous studies located in monsoonal area (Figure 1). Three tree ring $\delta^{18}O$ chronologies in northwest India, eastern Nepal and Bhutan were controlled by monsoon season rainfall or PDSI (Sano et al., 2011; Sano et al., 2013; Sano et al., submitted), and the two new tree ring $\delta^{18}O$ records obtained in the present study (JG and Ganesh) are negatively correlated with June-September PDSI in northern India. In addition, the five tree ring $\delta^{18}O$ records for the Himalaya region are significantly correlated each other during the common period (Table 2). These results indicate that five tree ring $\delta^{18}O$ records reflect a common controlling factor that may be related to regional climate. Therefore, we combined two tree ring $\delta^{18}O$ records in this study with three previously published tree ring $\delta^{18}O$ chronologies to construct a regional tree ring $\delta^{18}O$ record. The five $\delta^{18}O$ records were individually normalized over the interval from 1801-2000 CE, and then averaged to produce a regional Himalayan $\delta^{18}O$ record (H5 $\delta^{18}O$ record) for the entire interval (Figure 4f). Only one chronology (JG) spans an interval prior to 1742 CE, and therefore we focus on the interval from 1743-2008 CE in this study.

(3) The authors draw conclusions that may not be supported by their time series analysis methods. A section describing the spectral analysis methods, software, etc. that were used should be added to the methods section so that their results could be replicated by other researchers. Additionally, it is not clear how the authors determined the significance levels plotted in Figure 7 – this should be explained. A few additional comments/questions regarding the time series analysis are provided below: The conclusion that their record captures centennial-scale variability requires more justification considering their record is only 350 years long. They claim significant spectral power at 160 and 350 year intervals (Fig 7) – though the 350 year peak is the secular trend in their 350 year dataset, and the 160 year cycle may also not be significant-more details about how significance levels are determined should be provided. It was not clear why a 31-year moving correlation was used in figure 9 – could you expand on this choice?

**Answer**: Thank you for helpful suggestion. We have added the related sentence in Section 2.3 Meteorological data and climate analyses. We checked the codes that were used to calculate

the Power Spectra based on the multi-taper method in previous manuscript. The calculation of the confidence level in low frequency have some problems. We recalculated power spectra based on the multi-taper method using the Software "kSpectra Toolkit"(v3.4). The results show that H5 regional tree ring $\delta^{18}$O record contains several high-frequency periodicities (4 and 5 years), as well as lower frequency periodicities (~133 years) (New Figure 7).

[Figure]

Figure Caption: Figure 7: Multi-taper power spectra for the H5 regional tree ring $\delta^{18}$O record.

*3.3 Interannual variability of the ISM inferred from the regional tree ring $\delta^{18}$O record*

The results of spectral analysis using the multi-taper method (Mann and Lees, 1996) indicates that the H5 regional tree ring $\delta^{18}$O record contains several high-frequency quasi-periodicities (4 and 5 years), as well as lower frequency periodicities (~133 years) at a confidence level greater than 99% (Figure 7).

Maybe our explanation on 31-year moving correlation is not so clear. ENSO-Monsoon teleconnection is a nonstationary process. We need to evaluate the relationship between ENSO and monsoon. Because ENSO has the strongest power in wave length or cycles 2-7 yr, 31-year or 21-year window moving correlation between ENSO and precipitation were used to evaluate the stability of relationship between ENSO and climate. 31-year and 21-year window can cover the main cycles (2-7 years) of ENSO. For example, Camberlin et al., (2004, Climate Dynamics) used 31-year moving correlations between NINO3 SST and seasonal rainfall anomalies over a

few regions to see ENSO/rainfall teleconnections. 21-year moving window correlation analysis between ENSO and climate was used to investigate the relationship between ENSO and East Asian winter monsoon/ precipitation and flood pulse in the Mekong River Basin (Kim et al., 2016; Räsänen and Kummu, 2013)

Camberlin P, Chauvin F, Douville H, et al. Simulated ENSO-tropical rainfall teleconnections in present-day and under enhanced greenhouse gases conditions. Climate Dynamics, 2004, 23(6):641-657.
Kim J W, An S I, Jun S Y, et al. ENSO and East Asian winter monsoon relationship modulation associated with the anomalous northwest Pacific anticyclone. Climate Dynamics, 2016:1-23.
Räsänen T A, Kummu M. Spatiotemporal influences of ENSO on precipitation and flood pulse in the Mekong River Basin. Journal of Hydrology, 2013, 476(1):154-168.

(4) Writing is imprecise and organization needs improvement. We have provided a few of the more pressing examples to help guide revisions:

- The methods section requires substantial additions. First, the time series analysis methods used need to be described in an additional subsection, and in enough detail that other researchers could recreate the analysis. Second, many of the paragraphs in the results start with a description of how an analysis was done - these should be moved to the methods section.

**Answer**: Thank you for helpful suggestions. We have revised this part according to your suggestions. Please see the revised part.

*2.3 Climate analyses and Statistical Analysis*

In the northern Indian subcontinent, the monsoon season is from June to September. The summer monsoon season supplies 78% and 83% of the annual precipitation for Kathmandu and New Delhi, respectively. The Indian monsoon index (IMI) (Wang et al., 2001), the intensity of monsoon circulation (Webster and Yang, 1992) and All India Rainfall (AIR, obtained from the Indian Institute of Tropical Meteorology , Pune, India) were selected as proxies for the Indian summer monsoon in order to investigate the relationship between tree ring cellulose $\delta^{18}O$ variations and the monsoon. In addition, we used the Royal Netherlands Meteorological Institute Climate Explorer (http://www.knmi.nl/) to determine spatial correlations between tree-ring cellulose $\delta^{18}O$, precipitation (GPCC V7) and sea-surface temperature (SST) values

obtained from the National Climatic Data Center v4 data set. Temperature reconstructions for the Indian Ocean (Tierney et al., 2015) and the Tibetan Plateau (Cook et al., 2013; Shi et al., 2015; Wang et al., 2015), spanning the last 400 years, were used to obtain a record of the history of land-ocean thermal contrast. "kSpectra Toolkit"(v3.4) was employed to calculate power spectrum of the regional tree ring oxygen isotope chronology. The 95% (±1.96σ) confidence limits for each chronology and the regional chronology were calculated to show the uncertainty of each chronology and the regional chronology, respectively (except for the tree-ring chronology from Hulma, western Nepal, because the chronology was produced by pooling method, and therefore the uncertainty of this chronology was not able to show).

- The introduction brings up several relevant factors about the ISM without relating them directly to this study. This section would be more impactful if it were better focused on what is known about how these factors influence the ISM rather than just providing a list. This addition would help clarify how your results improve our understanding of the ISM.

**Answer**: Thank you for helpful suggestions. We have revised this part according to your suggestions. Please see the revised Introduction.

**1 Introduction**

[revised manuscript text omitted]

- The discussion section analyzing why there may be a weakening monsoonal circulation over last few hundred years requires a more in-depth analysis. The presented tree ring records cannot answer this question, and the question diverges from the main focus of the paper. The land-sea contrast mechanism described is potentially interesting, but the authors need to be more descriptive about: (a) how well do we know there has been a change in the land-sea temperature contrast? (b) are there other potential explanations, given the long list of factors influencing the ISM the authors list in the introduction?

**Answer**: We have discussed possible reasons to result in the weakening Indian summer monsoon. Sun activity and atmospheric CO2 content were not responsible for the reduction of ISM. Decreased land-sea temperature contrast may be the main reason, based on the fact that low-frequency variations in our regional tree-ring chronology are well correlated with those in the land-sea contrast data. In addition, aerosol emissions may be another reason to cause weakened ISM. Please see the revised part.

[revised manuscript text omitted]

Following the SPATIAL laboratory group discussion, Rich Fiorella compiled this short comment based on input from Gabe Bowen, Rose Smith, Annie Putman, Crystal Tulley-Cordova, Chao Ma, Zhongyin Cai, Yusuf Jameel, Brenden Fischer-Femal, and Sagarika Banerjee.

---

## Author Comment (AC4) · 29 Apr 2017

**Referee #3**

In their work, Xu et al. develop two new tree-rings isotopic chronologies of $\delta^{18}O$ from Northern India, and, combined with three other $\delta^{18}O$ chronologies, propose a multi-decadal regional reconstruction of the Indian summer monsoon. The regional record is further investigated using correlation and spectral analyses to document: 1) the drivers of Indian summer monsoon variability, and 2) the long-term trends of Indian summer monsoon intensity.

The new Data and this regional reconstruction are valuable to document a key hydroclimate component of the region, which lacks high resolution and long-term proxy records.

The methodology used in this paper as well as the results are robust. The data analyses, however, would benefit from further in depth exploration of each chronology signal. The discussion needs more regional scope but this can be improved if more data analyses are carried.

**General Comments**

- The authors present only correlations between the five $\delta^{18}O$ chronologies. How are the correlations for high and low frequency between the 5 chronologies?

**Answer:** Thanks for your helpful suggestions. We have added the correlation coefficient between the five $\delta^{18}O$ chronologies at multi-decadal time scale into Table 2 and the related sentences "the five tree ring $\delta^{18}O$ records for the Himalaya region are significantly correlated with each other at inter-annual time scale during the common period, and in most cases 31-year running averages of five tree ring $\delta^{18}O$ chronologies show significant correlations at multi-decadal time scale (Table 2)" in the manuscript. In general, the correlation coefficient among $\delta^{18}O$ chronologies decreased when the distances between two near $\delta^{18}O$ chronologies increased.

Table 2: Correlation coefficients between the tree ring $\delta^{18}O$ records from different sampling locations.

| r-annual | Manali | JG | Hulma | Ganesh |
|---|---|---|---|---|
| JG | 0.50* | | | |
| Hulma | 0.52* | 0.51* | | |
| Ganesh | 0.47* | 0.66* | 0.61* | |
| Wache | 0.23* | 0.26* | 0.37* | 0.52* |

| r-multi-decadal | Manali | JG | Hulma | Ganesh |
|---|---|---|---|---|
| JG | 0.36* | | | |
| Hulma | 0.37* | 0.64* | | |
| Ganesh | -0.03 | 0.94* | 0.66* | |
| Wache | 0.11 | 0.39* | 0.38* | 0.70* |

*$p<0.01$

- Climate correlations for each chronology δ[18]O are summarized in the text however a figure of correlation between each chronology and the main climate factor (precipitation and/or PDSI) should be presented to assess the climate signal in each chronology before creating the composite regional signal.

**Answer:** Thanks for your helpful suggestion. We have added the correlation between tree ring δ[18]O in five sites and regional June-September PDSI/Precipitation in Table 1. Because the correlations between tree ring δ[18]O in Manali, Hulma and Wache and monsoon season (JJAS) precipitation/PDSI were already showed and mechanisms that JJAS precipitation/PDSI affect tree ring δ[18]O in Manali, Hulma and Wache were already explained in previous study. (Sano et al., 2011, 2013, and submitted), we added climatic response of each tree ring δ[18]O chronology in Table 1.

Table 1. Tree ring cellulose oxygen isotope data sets used in this study

| No. | Sample ID | Location | Period | Tree species | Mean (1951-2000) | Climatic response of tree ring δ[18]O | Data source |
|---|---|---|---|---|---|---|---|
| 1 | Manali | 32°13′N, 77°13′E, 2700 masl, India | 1768-2008 | *Abies pindrow* | 29.97‰ | Regional JJAS PDSI $r$ =-0.67 | Sano et al., submitted |
| 2 | JG | 29°38′N, 79°51′E, 3849 masl, India | 1641-2008 | *Cedrus deodara* | 30.39‰ | Regional JJAS PDSI $r$ =-0.50 | This study |
| 3 | Hulma | 29°51′N, 81°56′E, 3850 masl, Nepal | 1778–2000 | *Abies spectabilis* | 25.94‰ | Regional JJAS PDSI $r$ =-0.73 | Sano et al., 2011 |
| 4 | Ganesh | 28°10′N, 85°11′E, 3550 masl, Nepal | 1801-2000 | *Abies spectabilis* | 23.01‰ | Regional JJAS PDSI $r$ =-0.55 | This study |
| 5 | Wache | 27°59′N, 90°00′E, 3500 masl, Bhutan | 1743-2011 | *Larix griffithii* | 19.38‰ | Regional JJAS Precipitation $r$ =-0.59 | Sano et al., 2013 |

- The composite signal (average of 5 centered δ[18]O chronologies) should be presented with a standard deviation or an uncertainty term in order to see if the uncertainty changed over time. This will be important to discuss the relationship between the composite regional signal and its relation with Indian summer monsoon indices for the last ~200 years.

**Answer:** Thanks for your helpful suggestion. We have added the 95% (±1.96σ) confidence limits to evaluate the uncertainty not only for the regional chronology but also for each tree ring δ[18]O chronology (except for chronology in Hulma, because the tree ring oxygen isotopes chronology was produced by pooling method, and therefore the uncertainty of this chronology was not able to show). Please see the following figure. gray shadows show the 95% (±1.96σ) confidence limits for each chronology.

[Figure]

Figure Caption: New Figure 4: Tree ring oxygen isotope chronologies from five sites (a-e) and the regional tree ring oxygen isotope chronology (f). (black line: mean values for all samples; red line: 31-year running average for the chronology; gray shadows: the 95% (±1.96σ) confidence limits)

- The $\delta^{18}O$ signal is often described from a theoretical perspective. Could the authors provide more evidence of the $\delta^{18}O$ signal in these particular chronologies? Comparison with RH or source water isotopes? Or using process based evaluation by means of $\delta^{18}O$ forward modelling (Eg. Evans 2007).

**Answer:** Source water $\delta^{18}O$ and relative humidity are two main controlling factor for tree ring cellulose $\delta^{18}O$. Comparison between tree ring $\delta^{18}O$ and source water $\delta^{18}O$/relative humidity and using tree ring $\delta^{18}O$ forward modelling would be helpful to understand source water $\delta^{18}O$ and relative humidity influences' on tree ring $\delta^{18}O$. However, long-term continuous precipitation $\delta^{18}O$ records are scarce in the study area. A better model parameterization for each species in each site depends on many observed parameters that are not available in five sites. The good things are that previous studies already showed the relationship between precipitation $\delta^{18}O$/relative humidity and tree ring $\delta^{18}O$ in study area. For example, Sano et al. (2011) indicated that tree ring $\delta^{18}O$ in Hulma showed negative correlations with June-September relative humidity and positive correlations with June-September precipitation $\delta^{18}O$ in New Delhi. Sano et al., (submitted) revealed that tree ring $\delta^{18}O$ in Manali showed the negative correlations with June-September relative humidity. In some sites, relative humidity record is not available. PDSI was employed to evaluate the relationship between tree ring $\delta^{18}O$ and moisture condition. Tree ring $\delta^{18}O$ in JG, Ganesh and Wache showed negative correlations with regional PDSI. Such negative correlations between tree ring $\delta^{18}O$ and summer PDSI or relative humidity have also been found in other areas of monsoonal Asia, such as northern Laos (Xu et al., 2013a), northern Vietnam (Sano et al., 2012), southeast Tibet Plateau (Grießinger et al., 2011; 2016, Liu et al., 2013, Wernicke et al., 2015), southeast China (Xu et al., 2013b, Xu et al., 2016) and Japan (Sakashita et al., 2015; Yamaguchi et al., 2010).

Grießinger J, Bräuning A, Helle G, Thomas A, Schleser G (2011) Late Holocene Asian summer monsoon variability reflected by $\delta^{18}O$ in tree-rings from Tibetan junipers. Geophysical Research Letters 38 (3):L03701

Grießinger J, Bräuning A, Helle G, Hochreuther P, Schleser G (2016) Late Holocene relative humidity history on the southeastern Tibetan plateau inferred from a tree-ring $\delta^{18}O$ record: Recent decrease and conditions during the last 1500 years. Quaternary International, In press

Liu X, Zeng X, Leavitt SW, Wang W, An W, Xu G, Sun W, Yu W, Qin D, Ren J (2013) A 400-year tree-ring $\delta^{18}O$ chronology for the southeastern Tibetan Plateau: Implications for inferring variations of the regional hydroclimate. Global & Planetary Change 104:23-33

Sakashita W, Yokoyama Y, Miyahara H, (2015) Relationship between early summer precipitation in Japan and the El Niño-Southern and Pacific Decadal Oscillations over the past 400 years. Quaternary International, 397(4):300-306.

Sano M, Xu C, Nakatsuka T (2012). A 300-year Vietnam hydroclimate and ENSO variability record reconstructed from tree ring $\delta^{18}O$. Journal of Geophysical Research Atmospheres, 117(D12):12115.

Wernicke J, Grießinger J, Hochreuther P, Bräuning A (2015) Variability of summer humidity during the past 800 years on the eastern Tibetan Plateau inferred from $\delta^{18}O$ of tree-ring cellulose. Climate of the Past11:327-337

Xu C, Sano M, Nakatsuka T(2013a). A 400-year record of hydroclimate variability and local ENSO history in northern Southeast Asia inferred from tree-ring $\delta^{18}O$.

Palaeogeography Palaeoclimatology Palaeoecology, 386:588-598.

Xu C, Zheng H, Nakatsuka T, Sano M (2013b) Oxygen isotope signatures preserved in tree ring cellulose as a proxy for April–September precipitation in Fujian, the subtropical region of southeast China. Journal of Geophysical Research: Atmospheres 118 (23):12,805-812,815

Xu C., J. Ge, T. Nakatsuka, L. Yi, H. Zheng, and M. Sano (2016), Potential utility of tree ring $\delta$18O series for reconstructing precipitation records from the lower reaches of the Yangtze River, southeast China, J. Geophys. Res. Atmos., 121, doi:10.1002/2015JD023610.

Yamaguchi Y T, Hughen K A. (2010) Synchronized Northern Hemisphere climate change and solar magnetic cycles during the Maunder Minimum. Proceedings of the National Academy of Sciences of the United States of America, 107(48):20697-20702.

- Analyses of observations would enhance the strength of the tree rings data: trends of rainfall for the region, as well as the various indices discussed in the paper. Additionally, $\delta^{18}O$ tree-rings and rainfall amount over the observations period should be plotted to strengthen the interpretation of the amount effect described in the results-discussion sections. This can be done for each site or at regional scale.

**Answer:** Thanks for your suggestion. We have added the description of nature of long-term variations in modern instrumental rainfall data based on the Sontakke et al., 2008 and Bhutiyani et al., 2010 in section 3.4 (*Centennial variability of the ISM inferred from the regional tree ring $\delta^{18}O$ record*). Please see the Section 3.4 (below).

[revised manuscript text omitted]

- In Page 9 paragraph 1 (~ line 5): Is the RH threshold 1%?

**Answer:** Yes, we refer to the results from Managave (2014). Maybe our explanation on relative humidity's influence on the correlation between two proxies is not so clear. We have revised this part. "Relative humidity has an important impact on tree ring $\delta^{18}O$ (Roden et al., 2000). Lower relative humidity result in enhanced evaporative enrichment of leaf water and then higher tree ring cellulose $\delta^{18}O$, while the relative humidity may not affect speleothem $\delta^{18}O$ when relative humidity does not correlate with precipitation $\delta^{18}O$ (Managave, 2014). Model results show that relative humidity's influences on the correlation between tree ring $\delta^{18}O$ and speleothem $\delta^{18}O$ is more pronounced in the regions where the variation of relative humidity during the growing season exceeds 1% (Managave, 2014)."

Managave, S. R.: Model evaluation of the coherence of a common source water oxygen isotopic signal recorded by tree-ring cellulose and speleothem calcite, Geochemistry Geophysics Geosystems, 15, 905–922, 2014.

- In Page 9 paragraph 1 (~ line 5): further discussion is required when assessing how the source $\delta^{18}O$ is integrated differently between tree-rings and speleothem proxies.

**Answer:** Thanks for your helpful suggestion. We have revised this part. Please see the revised part. "Based on the oxygen isotope fractionation theory, tree ring $\delta^{18}O$ and speleothem $\delta^{18}O$ should share similar changes (Managave, 2014) if both of them inherit a common source water $\delta^{18}O$ signal, as shown by Ramesh, et al (2013). The following reasons may cause incoherence between regional tree ring $\delta^{18}O$ and speleothem $\delta^{18}O$.

Other controlling factors differentially affect tree ring $\delta^{18}O$ and speleothem $\delta^{18}O$ values. Relative humidity has an important impact on tree ring $\delta^{18}O$ (Roden et al., 2000). Lower relative humidity result in enhanced evaporative enrichment of leaf water and then higher tree ring cellulose $\delta^{18}O$, while the relative humidity may not affect speleothem $\delta^{18}O$ when relative humidity does not correlate with precipitation $\delta^{18}O$ (Managave, 2014). Model results show that relative humidity's influences on the correlation between tree ring $\delta^{18}O$ and speleothem $\delta^{18}O$ is more pronounced in the regions where the variation of relative humidity during the growing season exceeds 1% (Managave, 2014). In contrast, the cave epikarst dynamics affect speleothems $\delta^{18}O$ significantly (Lachniet, 2009). The infiltrating water from different rainfall events may be stored and mixed in the epikarst. Lag times of $\delta^{18}O$ values in drip waters relative to rainfall are several years or decades in some locations (Lachniet, 2009), and a slow transit time smoothed climate signal. These processes may result in different source water for tree ring and speleothem. In addition, limited three $^{230}Th$ dates points (3 control points) and relative large age uncertainty (9-31 years) of speleothems $\delta^{18}O$ time series during the common period of 1743-2000 may result in the incoherence between tree ring and speleothems $\delta^{18}O$. Long-term process-based study on tree ring $\delta^{18}O$ and speleothem $\delta^{18}O$ variations in future study are needed for a better understanding for climatic implication of two proxies."

- Page 10 from line 10 to 25: the text needs substantial editing, there are lot of repetitions and the discussion is not clear. Often the authors start describing the implication of their record and its comparison with regional records without an in-depth discussion.

**Answer:** Thanks for your helpful suggestion. We have described the implication of our records and comparison with other records in the study in previous paragraphs. In this paragraph, we try to explain the reason that caused the weakened ISM. We revised this paragraph. Please see the revised part. "The land-sea thermal contrast which is also an important influencing factor for ISM (Roxy et al., 2015), is evaluated by atmospheric temperature gradient between the Tibetan Plateau and the tropical Indian Ocean (Fu and Fletcher, 1985; Sun et al., 2010). The history of land-sea thermal contrasts is reconstructed based on temperature differences between the Tibetan Plateau and the Indian Ocean (Figure 10a), and centennial variations of land-sea thermal contrasts are shown in Figure 10b. Three reconstructed land-sea thermal contrasts showed a decreasing trend since 1800 CE and 1820 CE (Figure 10b), and the H5 record exhibits a similar pattern of changes on a centennial scale (Figure 10c). The decreasing land-sea thermal contrast since 1800 and 1820 CE has resulted in a weaker ISM, and the increasing trend of the H5 record since 1820 CE also indicates a reduced ISM intensity. In addition, aerosol emissions may be another reason to cause weakened ISM. Because the aerosol-induced differential cooling of the source and nonsource regions resulted in not only reduced local land-ocean surface thermal contrast but also weaken large-scale meridional atmospheric temperature gradients, both of which caused weakening Indian summer monsoon circulation (Bollasina et al., 2011; Cowan and Cai, 2011). Long-term aerosol emissions record is needed to evaluate aerosol emission's influences on ISM in the past."

- Page 9 starting line ~20. The discussion here is interesting, however, needs some clarification and editing.

**Answer:** Thanks for your helpful suggestion. We have revised this paragraph according

to your suggestion. Please see the revised part. "However, in contrast, marine sediment records from the Western and Southeastern Arabian Sea exhibit an increasing trend of ISM strength over the last four centuries (Anderson et al., 2002; Chauhan et al., 2010). A recent study indicated that the contrasting trends in the ISM during the last several hundred years observed in geological records resulted from the different behavior of the Bay of Bengal branch and Arabian Sea branch of the ISM (Tan et al., 2016), and the Bay of Bengal branch of ISM weakened while intensity of Arabian Sea branch of the ISM increased during the last 200 years. However, the tree ring $\delta^{18}$O record in northwest India, influenced by the Arabian Sea branch of the ISM, exhibits a drying trend since 1950 CE (Sano et al., submitted), which does not support the idea of a strengthening Arabian Sea branch of the ISM (Anderson et al., 2002). Moreover, there are no calibrated radiocarbon dates for the last 300 years for the two records from the Arabian Sea (Anderson et al., 2002a; Chauhan et al., 2010). We suggest that further high-resolution and well-dated ISM records from western India are needed to improve our understanding of the behavior of the ISM. Although reconstructed All India monsoon rainfall does not show a significant decreasing trend during the period of 1813-2005 (Sontakke et al., 2008), the data from only four stations extend back to 1826 CE and four longest stations locate in central or southern India. Monsoon season drying trend in northern India revealed by H5 regional tree ring $\delta^{18}$O record may indicate that inland areas appear to be particularly sensitive to the weakening of monsoon circulation."

For the clarification: 1) the authors report a decreasing δ18O trend from 1743-1820. How many chronologies are included in this part of the record? According to Table 1 and Figure 4, only 2 chronologies extend back to 1743. Authors should use caution when making regional trend interpretations

**Answer:** Thanks for your helpful suggestion. We have added the sample depth in Figure. Please see the revised Figure 4 in Page 3 of this file. There are three tree ring $\delta^{18}$O chronologies since 1767 CE. The main conclusion is based on the result since 1800 CE. There are five tree ring $\delta^{18}$O chronologies since 1800 CE.

2) an increasing δ18O trend from 1820 to 2000 is observed in the δ18O tree-rings interpreted as an increase in the Indian monsoon intensity, also observed from other regional proxies. Is this trend also observed when considering the chronologies individually? What are the statistics for this trend?

**Answer:** an increasing tree ring $\delta^{18}$O trend from 1820 to 2000 is interpreted as a decrease in the Indian monsoon intensity. Such increasing trend of tree ring $\delta^{18}$O from 1820 to 2000 is also found in JG, Hulma, Ganesh and Wache (Please see Figure in Page in this file). The increased trend of regional tree ring $\delta^{18}$O chronology during the period of 1820-2008 was tested using linear regression. Please see the following figure and Figure in Page 11 in this file.

[Figure]

- Page 10, line 20. The discussion of factors (for instance aerosols) other than the decreasing land-ocean thermal contrast and their role in the decreasing Indian Monsoon intensity needs to be more detailed. What is the land-ocean thermal contrast resolution, interannual? Decadal? From Figure 11 the proxy records for ocean and land temperature do not seem to have the same temporal resolution.

**Answer:** Thanks for your suggestion. We have added the sentences on how the aerosols affect Indian summer monsoon. Please see the following paragraph. "Because the aerosol-induced differential cooling of the source and nonsource regions resulted in not only reduced local land-ocean surface thermal contrast but also weaken large-scale meridional atmospheric temperature gradients, both of which caused weakening Indian summer monsoon circulation (Bollasina et al., 2011; Cowan and Cai, 2011)."

For the land-ocean thermal contrast, temperature reconstruction in Tibet Plateau (Shi et al., 2015) represents a 10-year moving average and the resolution of SST reconstruction is annual in previous manuscript. In revised manuscript, we have added two annual-resolution summer temperature reconstruction in Tibetan Plateau (Cook et al., 2013; Wang et al., 2015) to evaluate the land-ocean thermal contrast history. Please see the following figure.

[Figure]

Figure Caption: Figure 10. a: Land-sea Temperature Anomaly based on three summer temperature reconstruction for the Tibetan Plateau and one Indian Ocean SST reconstruction; b and c: centennial variations of land-sea thermal contrasts and the H5 regional tree ring $\delta^{18}O$ chronology.

- when assessing the ENSO-and H5 regional $\delta^{18}O$ correlations, a 31-year window is too large (ENSO is 2-7 years). It would be helpful to investigate ENSO- and Tree-rings $\delta^{18}O$ correlation for individual chronologies to test whether the decorrelation is observed for all chronologies which reflect sites under slightly different precipitation

regime and Indian summer monsoon influence (based on Fig 1).

**Answer:** ENSO-Monsoon teleconnection is known to be a nonstationary process. We need to evaluate how the relationships between ENSO and monsoon changed in the past. Because ENSO has the strongest power in wave length or cycles 2-7 yr, 31-year or 21-year window moving correlations between ENSO and precipitation were used to evaluate the stability of relationship between ENSO and climate. The 31-year and 21-year window can cover the main cycles (2-7 years) of ENSO. For example, Camberlin et al., (2004, Climate Dynamics) used 31-year moving correlations between NINO3 SST and seasonal rainfall anomalies over a few regions to see ENSO/rainfall teleconnections. 21-year moving window correlation analysis between ENSO and climate was used to investigate the relationship between ENSO and East Asian winter monsoon/precipitation and flood pulse in the Mekong River Basin (Kim et al., 2016; Räsänen and Kummu, 2013)

Thanks for your helpful suggestion on the relationship between individual tree ring $\delta^{18}O$ chronology and ENSO. We did not add the related part in previous manuscript based on two reasons. 1) this manuscript mainly focused on the variations of ISM at regional scale rather than local scale. 2) Due to the complexity of ENSO-monsoon relationship, more tree ring $\delta^{18}O$ chronologies from Asian monsoon area (for example, Laos, Myanmar, Thailand, Vietnam, China and Japan) are needed. Now we are working on ENSO-Asian summer monsoon relationship based on lots of tree ring chronologies from Asian monsoon area.

Camberlin P, Chauvin F, Douville H, et al. Simulated ENSO-tropical rainfall teleconnections in present-day and under enhanced greenhouse gases conditions. Climate Dynamics, 2004, 23(6):641-657.
Kim J W, An S I, Jun S Y, et al. ENSO and East Asian winter monsoon relationship modulation associated with the anomalous northwest Pacific anticyclone. Climate Dynamics, 2016:1-23.
Räsänen T A, Kummu M. Spatiotemporal influences of ENSO on precipitation and flood pulse in the Mekong River Basin. Journal of Hydrology, 2013, 476(1):154-168.

**Specific comments**
- In Results and Discussion, section 3.1. The standard deviation of individual tree-ring cores can be added next to the mean.

**Answer:** Thanks for your suggestion. We have modified the section 3.1 according to your suggestion. Please see the revised part. "The oxygen isotopes of four individuals of *Abies spectabilis* in Ganesh (GE, central Nepal) and three individuals of *Cedrus deodara* in Jageshwar (JG, northern India) were measured for the interval from 1801-2000 CE and 1643-2008 CE, respectively. Individual tree ring $\delta^{18}O$ time series from four cores from central Nepal are shown in Figure 2a. The mean values (standard deviations) of the $\delta^{18}O$ time series from 224c, 233b, 235b, and 226a are 23.09‰(1.22‰), 22.66‰(1.27‰), 21.87‰(1.12‰), and 22.94‰(1.42‰), respectively, from 1901-2000 CE. The inter-tree differences in $\delta^{18}O$ values are small. The $\delta^{18}O$ values of the four cores exhibit peaks in 1813. The mean inter-series correlations (Rbar) among the cores range from 0.56-0.78 (Figure 2c), based on a 50-year window over the interval from 1801-2000 CE.

Three tree ring $\delta^{18}$O time series from northern India (JG) are shown in Figure 3a. The mean values (standard deviations) of the $\delta^{18}$O time series from 101c, 102c, and 103a are 30.11‰(1.49‰), 29.7‰(1.62‰) and 29.47‰(1.53‰), respectively, over the interval from 1694-2008 CE. Three tree ring $\delta^{18}$O time series in JG exhibit a consistent pattern of variations. The mean inter-series correlations (Rbar) among the cores range from 0.61-0.78 (Figure 3c), based on a 50-year window over the interval from 1641-2008 CE."

- In Table 1 the mean $\delta^{18}$O for each chronology should have a unit (‰)

**Answer:** We have modified the Table 1 according to your suggestion. Please see the following Figure.

Table 1. Tree ring cellulose oxygen isotope data sets used in this study

| No. | Sample ID | Location | Period | Tree species | Mean (1951-2000) | Climatic response of tree ring $\delta^{18}$O | Data source |
|---|---|---|---|---|---|---|---|
| 1 | Manali | 32°13′N, 77°13′E, 2700 masl, India | 1768-2008 | *Abies pindrow* | 29.97‰ | Regional JJAS PDSI r =-0.67 | Sano et al., submitted |
| 2 | JG | 29°38′N, 79°51′E, 3849 masl, India | 1641-2008 | *Cedrus deodara* | 30.39‰ | Regional JJAS PDSI r =-0.50 | This study |
| 3 | Hulma | 29°51′N, 81°56′E, 3850 masl, Nepal | 1778–2000 | *Abies spectabilis* | 25.94‰ | Regional JJAS PDSI r =-0.73 | Sano et al., 2011 |
| 4 | Ganesh | 28°10′N, 85°11′E, 3550 masl, Nepal | 1801-2000 | *Abies spectabilis* | 23.01‰ | Regional JJAS PDSI r =-0.55 | This study |
| 5 | Wache | 27°59′N, 90°00′E, 3500 masl, Bhutan | 1743-2011 | *Larix griffithii* | 19.38‰ | Regional JJAS PDSI r =-0.59 | Sano et al., 2013 |

- In Fig 1 and Table 2 the chronologies from previous studies have a different name. Bhutan in -Table 2 and Wache in Fig 1.

**Answer:** We have modified the Table 2 according to your suggestion. Please see the following Table 2.

Table 2: Correlation coefficients between the tree ring $\delta^{18}$O records from different sampling locations.

| *r-annual* | Manali | JG | Hulma | Ganesh |
|---|---|---|---|---|
| JG | 0.50* | | | |
| Hulma | 0.52* | 0.51* | | |
| Ganesh | 0.47* | 0.66* | 0.61* | |
| Wache | 0.23* | 0.26* | 0.37* | 0.52* |

| *r-multi-decadal* | Manali | JG | Hulma | Ganesh |
|---|---|---|---|---|
| JG | 0.36* | | | |
| Hulma | 0.37* | 0.64* | | |
| Ganesh | -0.03 | 0.94* | 0.66* | |
| Wache | 0.11 | 0.39* | 0.38* | 0.70* |

*$p$<0.01

- In Fig 2 the triangle and circle symbols have no legend.

**Answer:** We have modified the Figure according to your suggestion. Please see the following Figure.

[Figure]

- In Fig 3 the triangle and circle symbols legend is too big. This can be reduced to only the symbols without the line (same should be applied to Fig 2).

**Answer:** We have modified the Figure according to your suggestion. Please see the following Figure.

[Figure]

- In Fig 4, the sample depth (number) should be added in a graph below the composite time-series since the number of averaged trees over time is not the same (prior to 1740 and after 2000).

**Answer:** We have added the sample depth in the Figure according to your suggestion. Please see the following Figure.

[Figure]

- In Fig 5, add the location of all the sites to help visualize the strength of the field correlations.

**Answer:** We have modified the Figure according to your suggestion. Please see the following Figure.

[Figure]

- In Fig 8, add the location of the 18O chronologies for spatial reference of the SST correlations.

**Answer:** We have modified the Figure according to your suggestion. Please see the following Figure.

[Figure]

-Page 8 line 1, a word or punctuation is missing after the parenthesis (eastern-Pacific el Nino).

**Answer:** We have modified the related part according to your suggestion. Please see the revised part. "Most proxy-based ENSO reconstructions focused on canonical El Niño events (eastern-Pacific El Niño) that are characterized by unusually warm sea surface temperatures (SST) in the eastern equatorial Pacific (Gergis and Fowler, 2009; Li et al., 2011; McGregor et al., 2010)".

---

## Short Comment (SC2) · 2 May 2017

The PAGES Data Stewardship Integrative Activity seeks to advance best practices for sharing data generated and assembled as part of all PAGES-related activities. As part of this activity, a team of reviewers has been constituted for the "Climate of the Past 2000 years" Special Issue. The data team is reviewing the data handling within each of the CP-Discussion papers in relation to the CP data policy and current best practices. The team has identified essential and recommended additions for each paper, with the goal of achieving a high and consistent level of data stewardship across the 2k Special

Issue. We recognize that an additional effort will likely be required to meet the high level of data stewardship envisaged, and we appreciate dedication and contribution of the authors. This includes the use of Data Citations (see example in supplement). We ask authors to respond to our comments as part of the regular open interactive discussion. If you have any questions about PAGES Data Stewardship principles, please contact any of us directly.

Best wishes for the success of your paper,

2k Special Issue Data Review Team (Darrell Kaufman, Nerilie Abram, Belen Martrat, Raphael Neukom, Scott St. George) and ex-officio team members (Marie-France Loutre, Lucien von Gunten)

Essential additions for this paper:

(1) Add a "data availability" section that describes where the data can be accessed, including a Data Citation for the new data generated in this study (see below).

(2) Add Data Citations for each of the five datasets listed in Table 1, including both the previously published data, and the new data from this study. Note that the publication citation for record #3 is incorrect; a journal issue was assigned in 2012 (not 2011; doi:10.1177/0959683611430338).

(3) Add a note to explain that the spelling of the site name used in the previous paper is "Julma" rather than "Hulma" as it appears in the current paper.

(4) For those data not already in a public repository, submit essential metadata along with the time series shown in Figs 2a, 3a, and 4a, plus the averaged time series (H5) in Fig 4b, and its smoothed versions (Fig 10 (red) and Fig 11b (red)).

Recommended additions:

(1) Add Data Citations for each time series used to compare with the 18O tree-ring time series, including: Fig 5a (Indian rainfall); Fig 5b (Indian Monsoon); Fig 9 (ENSO

from McGregor and Wilson); Fig 10 (Stalagmite 18O); Fig 11 (Tibetan temperature and Indian Ocean SST)

(2) Submit for archival: (a) the correlation time series in Fig 9 and (b) the land-sea thermal contrast time series in Fig 11b (black).

Please also note the supplement to this comment:
http://www.clim-past-discuss.net/cp-2016-132/cp-2016-132-SC2-supplement.pdf

———————————————————

---

## Author Comment (AC5) · 6 Jun 2017

The PAGES Data Stewardship Integrative Activity seeks to advance best practices for sharing data generated and assembled as part of all PAGES-related activities. As part of this activity, a team of reviewers has been constituted for the "Climate of the Past 2000 years" Special Issue. The data team is reviewing the data handling within each of the CP-Discussion papers in relation to the CP data policy and current best practices. The team has identified essential and recommended additions for each paper, with the goal of achieving a high and consistent level of data stewardship across the 2k Special Issue. We recognize that an additional effort will likely be required to meet the high level of data stewardship envisaged, and we appreciate dedication and contribution of the authors. This includes the use of Data Citations (see example in supplement). We ask authors to respond to our comments as part of the regular open interactive discussion. If you have any questions about PAGES Data Stewardship principles, please contact any of us directly.

Best wishes for the success of your paper,

2k Special Issue Data Review Team (Darrell Kaufman, Nerilie Abram, Belen Martrat, Raphael Neukom, Scott St. George) and ex-officio team members (Marie-France Loutre, Lucien von Gunten)

(1) Add a "data availability" section that describes where the data can be accessed, including a Data Citation for the new data generated in this study (see below).

**Answer:** Five tree ring cellulose oxygen isotopes chronologies in this manuscript were not in a public repository, except for the data from Bhutan (available at DOI: 10.1002/jgrd.50664). This manuscript including two newly developed tree ring cellulose oxygen isotopes chronologies (JG and Ganesh) and another manuscript (Sano et al., submitted) including one tree ring cellulose oxygen isotopes chronology (Manali), which is also used in this manuscript, are still under review. Usually, these data should be open after the process of peer review. Therefore, we plan to contribute the data to NOAA Paleoclimatology Datasets (https://www.ncdc.noaa.gov/data-access/paleoclimatology-data) after both the manuscripts are published. We have added the description on the data availability in the Acknowledgments (red parts).

Acknowledgments:
This work was jointly funded by the Ministry of Science and Technology of the People's Republic of China (Grant No. 2016YFA0600502), the Chinese Academy of Sciences (CAS) Pioneer Hundred Talents Program, the National Natural Science Foundation of China (Grant No. 41672179, 41630529 and 41430531), an environmental research grant from the Sumitomo Foundation, Japan, a research grant from the Research Institute of Humanity and Nature, Kyoto, Japan, and grant in-aid from the Japan Society for the Promotion of Science Fellows (23242047 and 23-10262). Indian Space Research Organization's Geosphere Biosphere Programme supported RR and APD. This study was conducted in the framework of the Past Global Changes (PAGES) Asia2k programme. The tree ring cellulose oxygen isotope

data in this paper are available from the authors upon request (cxxu@mail.iggcas.ac.cn and msano@aoni.waseda.jp) and NOAA Paleoclimatology Datasets (https://www.ncdc.noaa.gov/data-access/paleoclimatology-data). We deeply appreciate the helpful comments from three anonymous reviewers and the group members of SPATIAL laboratory at the University of Utah to improve the manuscript.

(2) Add Data Citations for each of the five datasets listed in Table 1, including both the previously published data, and the new data from this study. Note that the publication citation for record #3 is incorrect; a journal issue was assigned in 2012 (not 2011; doi:10.1177/0959683611430338).

**Answer:** Thanks for your suggestions. We have modified the Table 1 according to the suggestions.

Table 1. Tree ring cellulose oxygen isotope data sets used in this study

| No. | Sample ID | Location | Period | Tree species | Mean | Climatic response of tree ring $\delta^{18}O$ | Data source |
|---|---|---|---|---|---|---|---|
| 1 | Manali | 32°13′N, 77°13′E, 2700 masl, India | 1768-2008 | *Abies pindrow* | 29.97‰ | Regional JJAS PDSI $r$ =-0.67 | Sano et al., submitted |
| 2 | JG | 29°38′N, 79°51′E, 3849 masl, India | 1641-2008 | *Cedrus deodara* | 30.39‰ | Regional JJAS PDSI $r$ =-0.50 | This study |
| 3 | Hulma | 29°51′N, 81°56′E, 3850 masl, Nepal | 1778–2000 | *Abies spectabilis* | 25.94‰ | Regional JJAS PDSI $r$ =-0.73 | Sano et al., 2012 |
| 4 | Ganesh | 28°10′N, 85°11′E, 3550 masl, Nepal | 1801-2000 | *Abies spectabilis* | 23.01‰ | Regional JJAS PDSI $r$ =-0.55 | This study |
| 5 | Wache | 27°59′N, 90°00′E, 3500 masl, Bhutan | 1743-2011 | *Larix griffithii* | 19.38‰ | Regional JJAS PDSI $r$ =-0.59 | Sano et al., 2013 |

(3) Add a note to explain that the spelling of the site name used in the previous paper is "Julma" rather than "Hulma" as it appears in the current paper.

**Answer:** In the previous paper (Sano et al., 2012), Hulma is the name for sampling site, while Julma is the name of meteorological station.

(4) For those data not already in a public repository, submit essential metadata along with the time series shown in Figs 2a, 3a, and 4a, plus the averaged time series (H5) in Fig 4b, and its smoothed versions (Fig 10 (red) and Fig 11b (red)).

**Answer:** Metadata may be helpful for published data. For the unpublished data, Table 1 provided similar information (name, location, length, climate implication, data source, etc) with metadata. Anyway, we plan to submit the data to NOAA after the manuscript was published, it will contain necessary information.

Recommended additions:

(1) Add Data Citations for each time series used to compare with the 18O tree-ring time series, including: Fig 5a (Indian rainfall); Fig 5b (Indian Monsoon); Fig 9 (ENSO from McGregor and Wilson); Fig 10 (Stalagmite 18O); Fig 11 (Tibetan temperature and Indian Ocean SST)

**Answer:** We have added the data citations for these records. Please see the following part.

Wang, B., Wu, R., and Lau, K., Indian monsoon index, http://apdrc.soest.hawaii.edu/projects/monsoon/definition.html, 2001

Mooley, D., Parthasarathy, B., Kumar, K., Sontakke, N., Munot, A., and Kothawale, D. Indian Institute of Tropical Meteorology Homogeneous Indian Monthly Rainfall Data Sets (1871-2014), http://www.tropmet.res.in/static_page.php?page_id=53, 2016

McGregor, S., Timmermann, A., and Timm, O.: A unified proxy for ENSO and PDO variability since 1650, World Data Center for Paleoclimatology, https://www.ncdc.noaa.gov/paleo-search/study/8732, 2010

Wilson, R., Tudhope, A., Brohan, P., Briffa, K., Osborn, T., and Tett, S.: Coral-based Tropical Sea Surface Temperature Reconstruction, World Data Center for Paleoclimatology, https://www.ncdc.noaa.gov/paleo-search/study/6359, 2006.

Sinha, A., Kathayat, G., Cheng, H., Breitenbach, S. F. M., Berkelhammer, M., Mudelsee, M., Biswas, J., and Edwards, R. L.: Trends and oscillations in the Indian summer monsoon rainfall over the last two millennia, Supplementary Data 2, Nat Commun, 6, 2015.

Tierney, J., Abram, N., Anchukaitis, K., Evans, M., Cyril, G., Halimeda, K., and Saenger, C., PAGES Ocean2K 400 Year Coral Data and Tropical SST Reconstructions, World Data Center for Paleoclimatology, https://www.ncdc.noaa.gov/paleo-search/study/17955, 2015.

Shi, F., Ge, Q., Bao, Y., Li, J., Yang, F., Ljungqvist, F. C., Solomina, O., Nakatsuka, T., Wang, N., and Zhao, S.: Asian 1,100 Year Multiproxy Gridded Summer Temperature Reconstructions, World Data Center for Paleoclimatology, http://ncdc.noaa.gov/paleo/study/18635, 2015.

Cook, E., Krusic, P., Anchukaitis. K., Buckley, M., Nakatsuka, T., Sano, M., and PAGES Asia2k Members: Asia 1200 Year Gridded Summer Temperature Reconstructions, World Data Center for Paleoclimatology, https://www.ncdc.noaa.gov/paleo-search/study/19523, 2013.

Wang, J., Yang, B., and Ljungqvist, F.: Eastern Tibetan Plateau 1000 Year Summer Temperature Reconstruction, World Data Center for Paleoclimatology, https://www.ncdc.noaa.gov/paleo-search/study/20590, 2015.

(2) Submit for archival: (a) the correlation time series in Fig 9 and (b) the land-sea thermal contrast time series in Fig 11b (black).

**Answer:** The data are easily calculated using raw data of five tree ring cellulose oxygen isotope chronologies and other data in public repository. After we contribute tree ring oxygen isotope data to a public repository, other researchers can reproduce the data based on their own interests.

---

## Referee Report (RR1)

The revised manuscript by Chenxi Xu et al. has been improved relative to the initial submission. However, the manuscript still contains conclusions not supported by the analyses. In general, we suggest the authors focus on presenting only the analyses that persuasively demonstrate their proposed mechanisms of variability. If this cannot be done, we suggest they soften the conclusions that cannot be robustly substantiated. The following review contains three major concerns should be addressed before publication, as well as a short list of smaller issues. Based on the work required to make the suggested changes, we recommend an additional round of major revisions.

**Major Concerns**
(1) The authors' treatment of uncertainty has improved from the initial version (e.g., the inclusion of 95% CIs and the age measurement uncertainties in Fig. 11), but the majority of the conclusions in this paper rest upon signals that have not been demonstrated to be more than noise. Specific examples are below:
   a. What is the uncertainty on the difference presented in Figure 10b? These records seem to have been generated by subtracting one proxy from another without propagating the error. Therefore, it's impossible to determine if there's any trend in these data, or if one reconstruction is different from another. The authors need to demonstrate this conclusively in order to validate their mechanism.
   b. We find the relationship in Figure 11 difficult to interpret due to substantial uncertainties in the age model of the stalagmite oxygen record. Time errors in the speleothem record are presented, and are on the same order as the timescale of the signal of interest (10-30 year uncertainty is ~10% of the length of the record, and on the same order of magnitude as the timescale of interest of the analysis). Thus, we expect the analysis comparing multidecadal signals in the tree ring stack to those in the speleothem record to be very sensitive to the uncertainty in the speleothem record timescales. For this analysis to be convincing, the authors need to address the relationship between signal and error.
   c. Could the authors explain in more detail why error estimates are not available for the Hulma record? It is not clear why they are not available simply because these records were generated by a pooling method.

(2) We are not yet convinced that these 5 tree ring records should be stacked. Presumably we'd want to stack these records to reduce local noise associated with a coherent regional signal. For the following two reasons, we are not convinced that the 5 sites experience a coherent regional climatic signal. First, the authors' response to our initial review indicated that low correlation between two sites was expected because they are far apart – this would seem to undercut the argument for placing them into the same stack, as the climatic drivers operating on these two sites are likely to be different. Second, Fig. 6 shows that only 3 of the 5 tree ring sites fall in the region where the H5 d18O variation has a significant correlation with precipitation amount variation suggesting that there's not a coherent, single regional signal across this region. We are

concerned that by stacking these 5 sites (as opposed to possibly the three or four most westerly sites), the authors may be averaging signals from two separate hydroclimatic regions. We strongly suggest that the motivation for stacking be clarified and strengthened.

(3) The spectral analysis method description and presentation has been substantially improved. The revised power spectrum exhibits a clear, significant signal at periods of ~4 and ~5 years. However, we are not convinced of that the centennial-scale peak, which is reported as corresponding to a ~133-year cycle, is a signal of a centennial-scale cycle as opposed to a secular trend. Part of our concern derives from the mismatch between the timescale of the cycle and the location of the signal peak in Fig. 7. The peak of the ~133-year cycle should be between 0.01 and 0.005 cycles/year. Instead the peak of the signal occurs at a value below 0.005 cycles/year. Because we cannot be confident that centennial scale variability is preserved in H5, the discussion of centennial variability as a preservation of the ISM signal is poorly substantiated. Therefore, we recommend that the authors refocus their paper to interpreting their record with respect to ENSO primarily. We feel this suggestions is robust for the following reasons:
   a. There is significant, robust spectral power at ~4 and ~5 years, which is consistent with ENSO timescales.
   b. Spatial patterns shown of the correlation between SST and d18O looks like ENSO variability, with the strongest impact in the eastern and central tropical Pacific (Fig. 8).

**Minor Comments**
(1) Links for Ganesh, JG, and Manali datasets to the NOAA Paleoclimatology Archive do not work. Please update so we can replicate analysis.
(2) Organization could still use improvement. For example, many methods are described in the results and discussion section. We need these in the methods section so they can be fully evaluated. For example:
   a. Pooled method (there must be a way to assess certainty!)
   b. Bandpass filter for decadal and multidecadal trends
(3) Figure 10: add legend for different colored shaded area.
(4) Figure 7: label y axis "**Log** power" and x axis with units (cycle/year)

This comment was prepared following a SPATIAL laboratory group discussion. Rich Fiorella, Annie Putman, and Chao Ma compiled this short comment with additional input from Gabe Bowen, Zhongyin Cai, and Yusuf Jameel.

---

## Author Response (AR2)

MS No.: cp-2016-132

"Decreasing Indian summer monsoon in northern Indian sub-continent during the last 180 years: evidence from five tree cellulose oxygen isotope chronologies"

Dear Dr. Evans,

Thank you for your letter regarding the above-mentioned manuscript. We would like to thank the reviewers for their constructive comments, and we have made corrections accordingly (see details below). Each comment (*italic*) from the editor or the reviewers is followed by our responses.

Best regards,

Masaki Sano on behalf of all authors

*Comments from editor:*

*Although three reviewers have recommended acceptance and minor revisions (please do address their suggested revisions), a fourth reviewer has suggested major revisions (document:cp-2016-132-referee-report-3.pdf). I have reviewed these suggestions carefully, and I believe that making these revisions will result in a stronger and more important contribution. In particular, they ask you to consider: (2) justification for the stacked record, given Fig 6; (1abc) strength of conclusions relative to propagated error, especially for the low frequency interpretations (cited figures); and (3) giving greater weight and refocusing the paper's discussion and conclusions around the most-substantiated "ENSO" timescales and patterns evident in the results, making the discussion of the low-frequency variability more speculative, given the uncertainties in the results (cited figures). Therefore I am asking you to revise the paper once more in light of the specific requests from this reviewer.*

   **Answer:** We have modified the manuscript based on your suggestions. In summary, 1) Regional tree ring oxygen isotope chronology based on five records showed relatively higher correlations with All Indian rainfall and Indian summer monsoon index, as compared to correlations seen with other chronologies derived from 3 or 4 local chronologies. 2) We have added the uncertainty for the low frequency variability. 3) We have described that Indian Ocean SST may partly modulate the Indian summer

monsoon-ENSO relationship. The following figure showed that the increasing trend of tree ring $\delta^{18}O$ from 1820 to 2000 is significant, whereas we have soften the conclusions that the land-sea temperature contrast may be a driving force of the increasing trend in the regional tree-ring record.

[Figure]

*In response to comments from the data stewardship team on the Data Availiability Section, we have confirmed that the Hulma, Wache, and Manali d18O datasets are available from the NOAA/NCEI repository at the URLs noted in the paper. Please confirm that the JG and Ganesh d18O chronologies and the composite regional d18O chronology developed in the present manuscript are also available via the NOAA/NCEI repository: per Climate of the Past and PAGES2k Special Issue policies, these revisions are required prior to acceptance of the manuscript. Currently we see only "shell" urls for the JG and Ganesh datasets, and in your prior response, you have not indicated a url for the composite o18 chronology reported in the present manuscript. For these latter three datasets, please forward a copy of the data that*

were delivered to NOAA/NCEI. If these data have not yet been delivered to NOAA/NCEI, please cc: me on your email which delivers those data. The goal is to get eyes on the data and to make sure that they are on their way or already at the repository before the paper is officially published.

**Answer:** As we sent an email with ccing the editor, the data (five local tree-ring oxygen isotope records and one composite tree-ring oxygen isotope chronology) have been sent to a data manager of *the NOAA/NCEI repository*. The data will be immediately opened once this manuscript is accepted.

*Comments from review 1*

*This sentence 'Monsoon precipitation in northwestern India showed a significant decreasing trend during the period of 1866-2006 (Bhutiyani et al., 2010)' should be deleted due to the same setence appears in the below section.*

**Answer:** We have deleted the sentences based on your suggestions.

*Comments from review 2*

*1-In figure 4 caption, when adding the uncertainty (95% confidence interval), precise that for the regional scale, it was not derived from all chronologies (except for Hulma chronology).*

We have described the methodology to calculate the 95% confidence intervals in the manuscript. As you noted, the intervals cannot be provided for the Humla chronology because of single isotope data in a single year, which is also mentioned in the revised ms.

*2-in Section 3.4. paragraph 5: "Although reconstructed All India monsoon rainfall does not show a significant decreasing trend during the period of 1813-2005 (Sontakke et al., 2008), the data from only four stations extend back to 1826 CE and four longest stations locate in central or southern India". This sentence needs some editing, below is a suggestion:*

*"Although reconstructed All India monsoon rainfall does not show a significant decreasing trend during the period of 1813-2005 (Sontakke et al., 2008), only four stations have data extending back to 1826 CE, and are located in central or south India."*

*This will apply if the four longest records (extending back to 1826) are the 4 stations located in central or south India.*

**Answer:** We have modified the sentence based on your suggestions. Please see Page 12, lines 13-15.

*3-in Section 3.5. paragraph 5: When authors state" Model results", what model they are referring to? it is not clear whether it is the "theoretical 18O fractionation model" or "the regression/correlation model when comparing their data with speleothem 18O". I suggest to the authors to clarify this in the text.*

**Answer:** We have deleted Section 3.5 including the paragraph, in response to comments from reviewer #3.

*4- In general when using tree ring, it should be hyphenated (tree-ring) when used as an adjective. Example: tree-ring record, tree-ring data, tree-ring 18O. Please correct it in manuscript accordingly.*

**Answer:** We have modified the whole manuscript according to your suggestion.

*Comments from review 3*

*The revised manuscript by Chenxi Xu et al. has been improved relative to the initial submission. However, the manuscript still contains conclusions not supported by the analyses. In general, we suggest the authors focus on presenting only the analyses that persuasively demonstrate their proposed mechanisms of variability. If this cannot be done, we suggest they soften the conclusions that cannot be robustly substantiated. The following review contains three major concerns should be addressed before publication, as well as a short list of smaller issues. Based on the work required to make the suggested changes, we recommend an additional round of major revisions. Major Concerns (1) The authors' treatment of uncertainty has improved from the initial version (e.g., the inclusion of 95% CIs and the age measurement uncertainties in Fig. 11), but the majority of the conclusions in this paper rest upon signals that have not been demonstrated to be more than noise. Specific examples are below: a. What is the uncertainty on the difference presented in Figure 10b? These records seem to have been generated by subtracting one proxy from another without propagating the error. Therefore, it's impossible to determine if there's any trend in these data, or if one*

*reconstruction is different from another. The authors need to demonstrate this conclusively in order to validate their mechanism. b. We find the relationship in Figure 11 difficult to interpret due to substantial uncertainties in the age model of the stalagmite oxygen record. Time errors in the speleothem record are presented, and are on the same order as the timescale of the signal of interest (10-30 year uncertainty is ~10% of the length of the record, and on the same order of magnitude as the timescale of interest of the analysis). Thus, we expect the analysis comparing multidecadal signals in the tree ring stack to those in the speleothem record to be very sensitive to the uncertainty in the speleothem record timescales. For this analysis to be convincing, the authors need to address the relationship between signal and error. c. Could the authors explain in more detail why error estimates are not available for the Hulma record? It is not clear why they are not available simply because these records were generated by a pooling method.*

**Answer:** We have added the uncertainty to the long-term variations in the regional tree ring chronology and the land-ocean thermal contrast. Please see Figure 10 and related part in the modified manuscript.

Because uncertainty of the temperature reconstructions for the Indian Ocean (Tierney et al., 2015) and the Tibetan Plateau (Cook et al., 2013; Shi et al., 2015; Wang et al., 2015) was provided as RMSE in the previous studies, the uncertainty range for the land-ocean contrast was estimated as the sum of RMSE of land temperature and Indian Ocean SST for each year. Using a low-pass filter, we extracted the low-frequency signals together with uncertainties (>100 years) for the tree-ring oxygen isotope chronology and the land-ocean contrast.

There is firm evidence of a weakening land-ocean thermal gradient over South Asia, which has been reported using long-term observations and model experiments by Roxy et al (2015). Specifically, rapid warming in the Indian Ocean and relatively subdued warming over the Indian subcontinent both contribute to the weakening land-sea gradient, and thereby reduce amounts of precipitation over parts of South Asia (Roxy et al., 2015). Based on the reviewer's suggestion, we dully noted in the ms that "while the long-term trends are overall consistent between our regional tree-ring chronology and the land-ocean thermal contrasts, possible propagation of uncertainty for the long-term land-sea gradient data should be kept in mind for interpretation."

Based on the reviewer's comment, we entirely deleted the Section 3.5 based on the stalagmite data, because of large uncertainty of age control.

*(2) We are not yet convinced that these 5 tree ring records should be stacked. Presumably we'd want to stack these records to reduce local noise associated with a coherent regional signal. For the following two reasons, we are not convinced that the 5 sites experience a coherent regional climatic signal. First, the authors' response to our initial review indicated that low correlation between two sites was expected because they are far apart – this would seem to undercut the argument for placing them into the same stack, as the climatic drivers operating on these two sites are likely to be different. Second, Fig. 6 shows that only 3 of the 5 tree ring sites fall in the region where the H5 d18O variation has a significant correlation with precipitation amount variation suggesting that there's not a coherent, single regional signal across this region. We are concerned that by stacking these 5 sites (as opposed to possibly the three or four most westerly sites), the authors may be averaging signals from two separate hydroclimatic regions. We strongly suggest that the motivation for stacking be clarified and strengthened.*

**Answer:** We built up three regional tree ring oxygen isotopes chronologies based on all five records (H5), four records (except Wache in Bhutan, the eastern site, H4) and three records (except Wache in Bhutan and Ganesh in Nepal, H3). The correlation analysis between three regional tree ring oxygen isotopes chronologies (H5, H4, H3) with All Indian rainfall (AIR), Indian monsoon index (IMI) and Intensity of monsoon circulation (Webster and Yang Monsoon Index (WYM) was employed to check which chronology can capture more regional signal. The results were shown in the following table. In general, the correlation coefficients between H5 and regional climate index (AIR, IMI, WYM) are higher than those between H4, H3 and regional climate index in most cases, although the correlation coefficient between H3 and IMI is higher than the correlation coefficient between H4, H5 and IMI. Therefore, regional oxygen isotope chronology based five tree ring records can capture more monsoon-related climate signal, which is the reason that 5 tree ring records were stacked.

| r | Five tree ring oxygen isotopes records (H5) vs AIR | Four tree ring oxygen isotopes records (H4, excluding Wache in Bhutan) vs AIR | Three tree ring oxygen isotopes records (H3, excluding Wache in |
|---|---|---|---|

|  |  |  | Bhutan and Ganesh in Nepal) vs AIR |
| --- | --- | --- | --- |
| Correlation coefficient with All Indian Rainfall (AIR) | -0.498 (1871-2008) | -0.476 (1871-2008) | -0.456 (1871-2008) |
| Correlation coefficient with Indian monsoon index (IMI, Wang et al., 2001) | -0.454 (1948-2008) | -0.438 (1948-2008) | -0.465 (1948-2008) |
| Correlation coefficient with intensity of the monsoon circulation (WYM, (Webster and Yang, 1992) | -0.424 (1948-2008) | -0.365 (1948-2008) | -0.318 (1948-2008) |

*(3) The spectral analysis method description and presentation has been substantially improved. The revised power spectrum exhibits a clear, significant signal at periods of ~4 and ~5 years. However, we are not convinced of that the centennial-scale peak, which is reported as corresponding to a ~133-year cycle, is a signal of a centennial-scale cycle as opposed to a secular trend. Part of our concern derives from the mismatch between the timescale of the cycle and the location of the signal peak in Fig. 7. The peak of the ~133-year cycle should be between 0.01 and 0.005 cycles/year. Instead the peak of the signal occurs at a value below 0.005 cycles/year. Because we cannot be confident that centennial scale variability is preserved in H5, the discussion of centennial variability as a preservation of the ISM signal is poorly substantiated. Therefore, we recommend that the authors refocus their paper to interpreting their record with respect to ENSO primarily. We feel this suggestions is robust for the following reasons: a. There is significant, robust spectral power at ~4 and ~5 years, which is consistent with ENSO timescales. b. Spatial patterns shown of the correlation*

*between SST and d18O looks like ENSO variability, with the strongest impact in the eastern and central tropical Pacific (Fig. 8).*

**Answer:** The results from kSpectra showed that the frequencies of 0.0075188 (~133 years), 0.1992188 (~5 years), 0.2548828 (~4 years) are significant at the confidence of 99% for the H5 tree ring oxygen isotopes during the period of 1743-2008, which is shown in Figure 7. Low frequency signal is obvious from the Figure 7. Because we focused on the long-term changes of ISM, low-pass filter (100 years) was used to extract low frequency signal of ISM. We have added a part describing the ISM-ENSO relationship based on your suggestions.

*Minor Comments*

*(1) Links for Ganesh, JG, and Manali datasets to the NOAA Paleoclimatology Archive do not work. Please update so we can replicate analysis.*

**Answer:** We have already submitted the data to NOAA. The data will be fully opened once the manuscript is accepted.

*(2) Organization could still use improvement. For example, many methods are described in the results and discussion section. We need these in the methods section so they can be fully evaluated. For example: a. Pooled method (there must be a way to assess certainty!) b. Bandpass filter for decadal and multidecadal trends*

**Answer:** We have modified organization of the methods, and have added some descriptions in accordance with reviewer's suggestions.

*(3) Figure 10: add legend for different colored shaded area.*

**Answer:** We have added legend for colored shaded area in Figure 10.

*(4) Figure 7: label y axis "Log power" and x axis with units (cycle/year)*

**Answer:** We have revised the Figure 7 according the reviewers' suggestions.

*This comment was prepared following a SPATIAL laboratory group discussion. Rich Fiorella, Annie Putman, and Chao Ma compiled this short comment with additional input from Gabe Bowen, Zhongyin Cai, and Yusuf Jameel.*

*Comments from review 4*

*accepted as is*

**Decreasing Indian summer monsoon in northern Indian sub-continent during the last 180 years: evidence from five tree cellulose oxygen isotope chronologies**

Chenxi Xu[1], Masaki Sano[2,3], A. P. Dimri[4], Rengaswamy Ramesh[5,6], Takeshi Nakatsuka[2], Feng Shi[1], Zhengtang Guo[1,7,8]

1. Key Laboratory of Cenozoic Geology and Environment, Institute of Geology and Geophysics, Chinese Academy of Sciences, Beijing 100029, China
2. Research Institute for Humanity and Nature, 457-4 Motoyama, Kamigamo, Kita-ku, Kyoto, Japan
3. Faculty of Human Sciences, Waseda University, 2-579-15 Mikajima, Tokorozawa 359-1192, Japan
4. School of Environmental Sciences, Jawaharlal Nehru University, New Delhi, India
5. Geoscience Division, Physical Research Laboratory, Navrangpura, Ahmedabad 380009, India
6. School of Earth and Planetary Sciences, National Institute of Science Education and Research, Odisha 752050, India
7. CAS Center for Excellence in Tibetan Plateau Earth Sciences, Beijing 100101, China
8. University of Chinese Academy of Sciences, Beijing, China

*Correspondence to*: Masaki Sano, (msano@aoni.waseda.jp)

**Abstract.** We have constructed a regional tree-ring cellulose oxygen isotope ($\delta^{18}O$) record for the northern Indian sub-continent based on two new records from northern India and central Nepal and three published records from northwestern India, western Nepal and Bhutan. The record spans the common interval from 1743-2008 CE. Correlation analysis reveals that the record is significantly and negatively correlated with the three regional climatic indices: All India Rainfall ($r = -0.5$, $p < 0.001$, $n = 138$), Indian monsoon index ($r = -0.45$, $p < 0.001$, $n = 51$) and the intensity of monsoonal circulation ($r = -0.42$, $p < 0.001$, $n = 51$). The close relationship between tree-ring cellulose $\delta^{18}O$ and the Indian summer monsoon (ISM) can be explained by oxygen isotope fractionation mechanisms. Our results indicate that the regional tree-ring cellulose $\delta^{18}O$ record is suitable for reconstructing high-resolution changes in the ISM. The record exhibits significant inter-annual and long-term variations. Inter-annual changes are closely related to the El Niño-Southern Oscillation (ENSO), which indicates that the ISM was affected by ENSO in the past. However, the ISM-ENSO relationship was not consistent over time, and it may be partly modulated by Indian Ocean sea surface temperature (SST). Long-term changes in the regional tree-ring $\delta^{18}O$ record indicate a possible trend of weakened ISM intensity since 1820. Decreasing ISM activity is also observed in various high-resolution ISM records from southwest China and Southeast Asia, and may be the result of reduced land-ocean thermal contrasts since 1820 CE.
* * *
Margin edits (tracked changes):

[revised manuscript text omitted]

speleothems oxygen isotopes fractionation model, the correlation between simulated tree-ring and

speleothem $\delta^{18}O$ decreased rapidlyshow that relative humidity's influences on the correlation between tree

ringtree-ring $\delta^{18}O$ and speleothem $\delta^{18}O$ is more pronounced in the regions where the variation of relative

humidity during the growing season exceeds 1% (Managave, 2014). In contrast, the cave epikarst dynamics

affect speleothems $\delta^{18}O$ significantly (Lachniet, 2009). The infiltrating water from different rainfall events

may be stored and mixed in the epikarst. Lag times of $\delta^{18}O$ values in drip waters relative to rainfall are

several years or decades in some locations (Lachniet, 2009), and a slow transit time smoothed climate signal.

These processes may result in different source water for tree ringtree-ring and speleothem. In addition,

limited three $^{230}$Th dates points (3 control points) and relative large age uncertainty (9-31 years) of

speleothems $\delta^{18}$O time series during the common period of 1743-2000 and uncertainty of H5 regional treering $\delta^{18}$O chronology may result in the incoherence between tree ringtree-ring and speleothems $\delta^{18}$O. Longterm process-based study on tree ringtree-ring $\delta^{18}$O and speleothem $\delta^{18}$O variations in same sampling site

in future study are needed for a better understanding for climatic implication of two proxies.

| ページ 24: [5] 削除 | pilgrimto | 2018/01/07 21:33:00 |
|---|---|---|

Wilson, R., Tudhope, A., Brohan, P., Briffa, K., Osborn, T., and Tett, S.: Two-hundred-fifty years of

reconstructed and modeled tropical temperatures, Journal of Geophysical Research, 111, C10007, 2006.

Wilson, R., Tudhope, A., Brohan, P., Briffa, K., Osborn, T., and Tett, S.:  Coral-based Tropical Sea Surface

Temperature Reconstruction, World Data Center for Paleoclimatology, https://www.ncdc.noaa.gov/paleosearch/study/6359, 2006.

| ページ 31: [6] 削除 | unknown | 2017/12/26 17:01:00 |
|---|---|---|

Tree ring

| ページ 31: [6] 削除 | unknown | 2017/12/26 17:01:00 |
|---|---|---|

Tree ring

ページ 36: [7] 削除      Masaki Sano      2018/01/26 15:04:00

ページ 36: [8] 削除      unknown      2018/01/08 11:05:00

ページ 36: [9] 削除      Masaki Sano      2018/01/21 16:07:00

/

| ページ 38: [10] 削除 | Masaki Sano | 2018/01/26 10:26:00 |
|---|---|---|

sea

| ページ 38: [11] 削除 | unknown | 2017/12/26 17:01:00 |
|---|---|---|

tree ring

| ページ 38: [12] 削除 | Masaki Sano | 2018/01/21 16:06:00 |
|---|---|---|

[Figure]

[Figure]

Figure 11. Comparison between multi-decadal regional tree ringtree-ring $\delta^{18}O$ variations (red line) with stalagmite $\delta^{18}O$ changes (black line) in northern India. Rhombus with error indicates the [230]Th dates with uncertainty in stalagmite $\delta^{18}O$ chronology. Shades indicate uncertainty for regional tree-ring $\delta^{18}O$ chronology.

---

## Author Response (AR3)

MS No.: cp-2016-132

 "Decreasing Indian summer monsoon in northern Indian sub-continent during the last 180 years: evidence from five tree cellulose oxygen isotope chronologies"

Dear Dr. Evans,

Thank you for your letter regarding the above-mentioned manuscript. As you know, all the data used in this paper are now available at the NOAA/WDC database. We have corrected some typos and descriptions that were not clear.

Best regards,

Masaki Sano on behalf of all authors

[revised manuscript text omitted]

---

## Author Response (AR4)

MS No.: cp-2016-132

"Decreasing Indian summer monsoon in northern Indian sub-continent during the last 180 years: evidence from five tree cellulose oxygen isotope chronologies"

Dear Dr. Evans,

Thank you for your letter regarding the above-mentioned manuscript. We have changed wording "should be" to "is" in the Data Availability Section.

Best regards,

Masaki Sano on behalf of all authors

[revised manuscript text omitted]